# DriveWorld-VLA: Unified Latent-Space World Modeling with Vision–Language–Action for Autonomous Driving

**Feiyang Jia** [*1 2] **Lin Liu** [*1 2] **Ziying Song** [1] **Caiyan Jia** [†1] **Hangjun Ye** [2] **Xiaoshuai Hao** [†2] **Long Chen** [‡2]

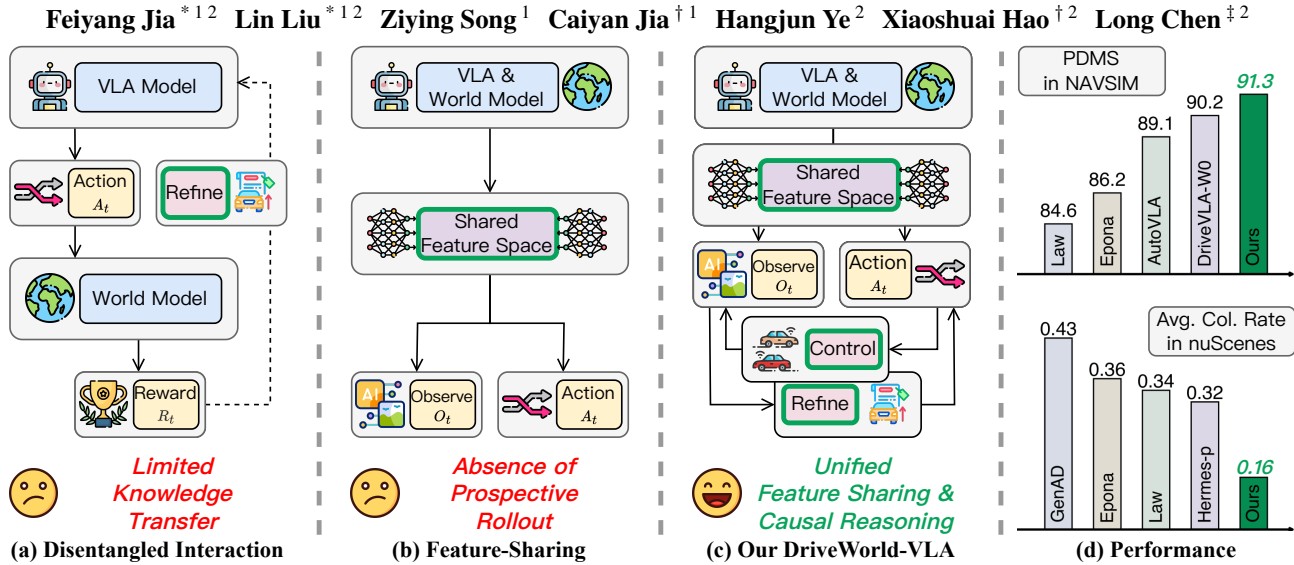

*Figure 1.* **Comparison of VLA & World Model Coupling Strategies.** *(a) Disentangled Interaction*: The world model acts as an external simulator, but its structural isolation from the VLA prevents effective latent knowledge transfer. *(b) Feature Sharing*: Despite sharing representations, these models lack action-conditioned causal reasoning, which limits their counterfactual imagination and long-horizon planning. *(c) Our DriveWorld-VLA*: By optimizing world model latents as decision variables, we enable unified causal "what-if" reasoning through controllable imagination in a shared latent space. *(d) Performance*: DriveWorld-VLA achieves SOTA results—91.3 PDMS on NAVSIMv1, 86.8 EPDMS on NAVSIMv2, and 0.16 CR on nuScenes—significantly outperforming specialized baselines like LAW (Li et al., 2024a), Epona (Zhang et al., 2025), and HERMES-p (Zhou et al., 2025b).

## Abstract

End-to-end (E2E) autonomous driving has recently attracted increasing interest in unifying Vision–Language–Action (VLA) with World Models to enhance decision-making and forward-looking imagination. However, existing methods fail to effectively unify future scene evolution and action planning within a single architecture due to inadequate sharing of latent states, limiting the impact of visual imagination on action decisions. To address this limitation, we propose **DriveWorld-VLA**, a novel framework that

unifies world modeling and planning within a latent space by tightly integrating VLA and world models at the representation level, which enables the VLA planner to benefit directly from holistic scene-evolution modeling and reducing reliance on dense annotated supervision. Additionally, **DriveWorld-VLA** incorporates the latent states of the world model as core decision-making states for the VLA planner, facilitating the planner to assess how candidate actions impact future scene evolution. By conducting world modeling entirely in the latent space, **DriveWorld-VLA** supports controllable, action-conditioned imagination at the feature level, avoiding expensive pixel-level rollouts. Extensive open-loop and closed-loop evaluations demonstrate the effectiveness of **DriveWorld-VLA**, which achieves state-of-the-art performance with 91.3 PDMS on NAVSIMv1, 86.8 EPDMS on NAVSIMv2, and 0.16 3-second average collision rate on nuScenes. Code and models are released at DriveWorld-VLA.

---

[*]Equal contribution , [†]Corresponding authors, [‡]Project leader. [1]School of Computer Science and Technology, Beijing Key Laboratory of Traffic Data Mining and Embodied Intelligence, Beijing Jiaotong University [2]Xiaomi EV. Correspondence to: Caiyan Jia <cyjia@bjtu.edu.cn>, Xiaoshuai Hao <haoxiaoshuai@xiaomi.com>.

*Proceedings of the 43rd International Conference on Machine Learning*, Seoul, South Korea. PMLR 306, 2026. Copyright 2026 by the author(s).

## 1. Introduction

Autonomous driving is undergoing a transformative paradigm shift from modular pipelines toward End-to-End (E2E) learning (Cui et al., 2024). Despite their success in mapping sensor inputs to control signals (Song et al., 2025a; Liu et al., 2025; Liao et al., 2025; Song et al., 2025b), these models often lack the capacity for long-horizon reasoning and fail to comprehend the causal consequences of their actions. In response, the integration of Vision-Language-Action (VLA) models with World Models (WM) has emerged as a promising frontier to bridge this gap. In this synergy, VLA models provide sophisticated multimodal perception and linguistic reasoning, while World Models empower agents with "prospective imagination" by explicitly modeling environmental dynamics and action-conditioned future states.

Despite this potential synergy, existing attempts to unify these paradigms remain loosely coupled, typically falling into two suboptimal categories. The first class includes *Disentangled Interaction* methods (Yan et al., 2025; Yang et al., 2025a), as shown in Figure 1.(a), which treat the world model merely as an external simulator or data source. However, these methods create a structural bottleneck that prevents the VLA from internalizing fundamental physical laws and environment dynamics. The second category is composed of *Feature-Sharing* approaches (Zhang et al., 2025; Li et al., 2025b; Zhou et al., 2025b), depicted in Figure 1.(b), which employ shared representations for joint prediction but fail to leverage the world model's most critical capability: explicit causal reasoning and prospective "what-if" imagination. Consequently, by neglecting the action-outcome causal chain, these models remain restricted to reactive planning rather than proactive, long-horizon optimization through counterfactual simulation.

To address these limitations, we propose *DriveWorld-VLA* (as shown in Figure 1.(c)), a unified framework that integrates VLA and World Models across both representation and decision-making through three core innovations. First, *Feature-Level Sharing* utilizes Large Language Model (LLM) hidden states as a shared latent space for both imagination and action prediction, enabling the model to internalize fundamental physical laws and environmental dynamics within its core reasoning engine. Second, *Action-Conditioned "What-If" Reasoning* leverages a Diffusion Transformer (DiT) architecture to perform prospective rollouts of multiple candidate actions; this empowers the agent to transition from reactive planning to proactive optimization by explicitly evaluating the long-term consequences of distinct trajectories. Third, a *Three-Stage Progressive Training paradigm* stabilizes joint optimization by sequentially aligning multi-modal perception with BEV generation, enforcing action-conditioned controllability via flow-

matching, and finally retroactively refining VLA decisions through a closed-loop reward mechanism. Extensive evaluations across open-loop and closed-loop benchmarks demonstrate that *DriveWorld-VLA* achieves state-of-the-art performance, significantly outperforming specialized baselines in both reasoning accuracy and safety-critical planning.

In summary, our main contributions are as follows:

- We propose *DriveWorld-VLA*, a tightly coupled framework where a world model serves as the reasoning engine bridging action and prospective imagination.
- We introduce Feature-Level Sharing that uses LLM hidden states as shared latent space for both imagination and action prediction, helping internalize physical laws and environment dynamics.
- We develop action-conditioned "what-if" reasoning to generate and evaluate candidate trajectories for proactive, consequence-aware decision-making.
- We conduct extensive evaluations on both closed-loop (NAVSIM) and open-loop (nuScenes) benchmarks, demonstrating feature-level sharing and causal rollout guidance are critical for robust decision-making.

## 2. Related Works

**World Models for Autonomous Driving** World models (WMs) have emerged as a cornerstone for autonomous driving, providing a technical foundation for modeling complex environmental evolutions (Jia et al., 2025; Tu et al., 2025; Guan et al., 2024; Feng et al., 2025b). Early efforts such as DriveWorld (Min et al., 2024) and Vista (Gao et al., 2024b) focus on generating spatiotemporally consistent video sequences, while LidarDM (Zyrianov et al., 2025) and Copilot4D (Zhang et al., 2023) explore driving-conditioned point cloud synthesis. Other works, including ViDAR (Yang et al., 2024) and Occ-LLM (Xu et al., 2025), treat occupancy prediction as a foundational representation for world dynamics. Beyond pixel-level generation, some models map observations into latent spaces to forecast agent behaviors and motion (Li et al., 2024a; Hu et al., 2022), partially bridging the gap between conventional pipelines and holistic traffic modeling. Despite these advances, the community lacks a unified objective, with research often split between high-fidelity future prediction and trajectory modeling. Only a few pioneering studies, such as HERMES (Zhou et al., 2025b) and Epona (Zhang et al., 2025), explicitly connect generative imagination with downstream planning. In this context, *DriveWorld-VLA* unifies these paradigms by integrating generative modeling and decision-making within a single framework.

**VLAs for Autonomous Driving** The rapid evolution of VLA models in autonomous driving is primarily fueled by the profound semantic understanding and reasoning capa-

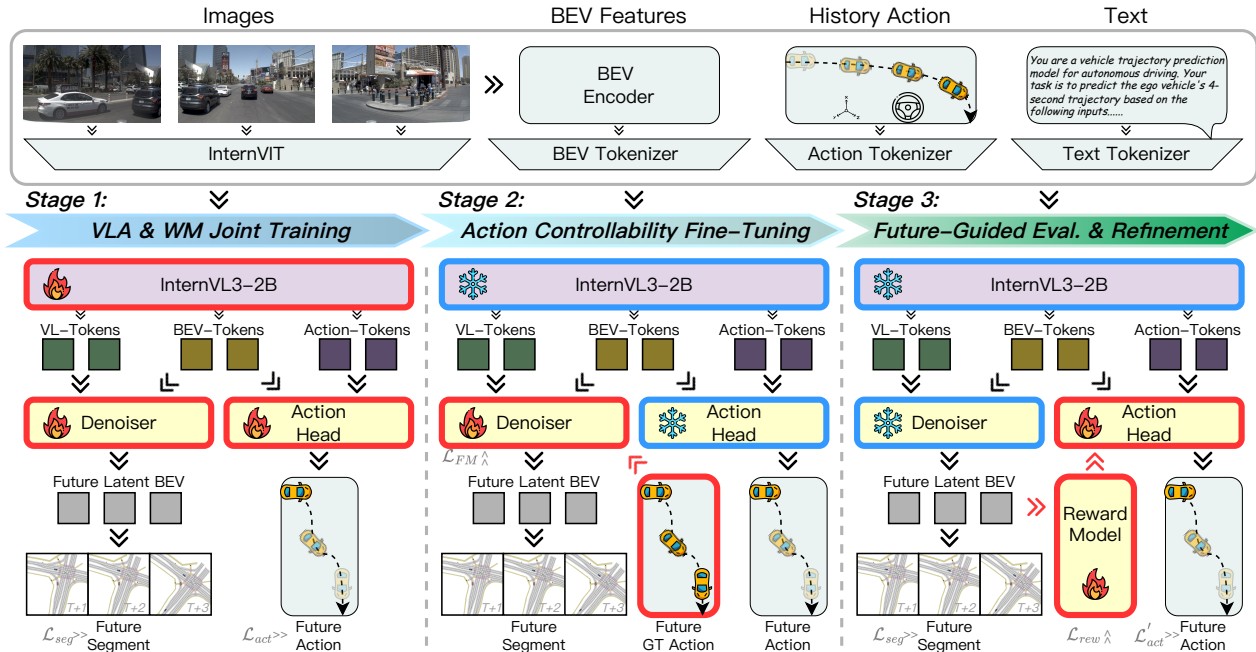

*Figure 2.* **DriveWorld-VLA pipeline.** DriveWorld-VLA unifies action and prospective imagination through a progressive training scheme. **Stage 1** jointly learns future BEV imagination and action prediction from a shared latent representation. **Stage 2** conditions the generative branch on future actions, enabling controllable imagination that maps a given action sequence to its corresponding future. **Stage 3** closes the loop: first predicts actions, then imagines the resulting future, and finally uses reward feedback to refine action prediction.

bilities of VLMs. Early research predominantly utilized VLMs as high-level semantic interpreters to process textualized driving logic and rule-based constraints (Xu et al., 2024; Zhou et al., 2025a; Yang et al., 2025b; Hao et al., 2025b;a). However, with the emergence of multi-modal foundation models, the paradigm has shifted toward end-to-end VLA architectures that directly map multi-sensor inputs and linguistic instructions to control trajectories (Shao et al., 2024; Renz et al., 2025; Jiang et al., 2025). Recent frontiers have explored the deep integration of VLAs with world models to enable future-aware reasoning and latent state prediction (Zhou et al., 2025b; Li et al., 2025b; Zhou et al., 2025c; Wang et al., 2025; Zheng et al., 2025; Gao et al., 2024a; Gu et al., 2024). For instance, HERMES (Zhou et al., 2025b) jointly models scene interpretation and evolution, while DriveVLA-W0 (Li et al., 2025b) employs future image prediction as a self-supervised signal for VLA refinement. *DriveWorld-VLA* advances this trajectory by unifying future visual prediction and action generation within a shared latent space, systematically bridging the gap between generative imagination and proactive decision-making.

## 3. Method

*DriveWorld-VLA* is designed to tightly integrate VLA model with World Model in a unified architecture that supports both multi-modal reasoning and prospective imagination. To progressively align representation learning, action con-

trollability, and consequence-aware decision making, we adopt a three-stage training paradigm. Each stage incrementally unlocks a key capability of the world model, while ensuring stable joint optimization with the VLA. Specifically, the training process is organized into three sequential stages: VLA & WM Joint Training, Action Controllability Fine-Tuning and Guided Evaluation & Refinement. Figure 2 shows the pipeline of *DriveWorld-VLA*.

### 3.1. VLA & WM Joint Training

*DriveWorld-VLA* supports multi-modal inputs, including multi-view images $\mathcal{I}_t$, textual prompts $\mathcal{T}_t$, historical actions $\mathcal{A}_{t-1}$, and BEV representations $\mathcal{B}_t$. All modalities are independently tokenized before being fed into the Vision-Language Model (VLM). Image and text tokenization follow InternVL (Zhu et al., 2025), while dedicated tokenizers are introduced for BEV features and historical actions. BEV features $\mathcal{B}_t \in \mathbb{R}^{H \times W \times C}$ are extracted by BEVFormer (Li et al., 2024c), flattened spatially, and projected into the VLM embedding space as BEV tokens. Historical ego actions are serialized into natural language prompts and concatenated with textual instructions, then encoded using the same text tokenizer as InternVL.

After tokenization, $\mathcal{I}_t$, $\mathcal{T}_t$, $\mathcal{A}_{t-1}$, and $\mathcal{B}_t$ are jointly fed into the VLM. The VLM aggregates information across all modalities and produces a sequence of hidden states. We extract the hidden states from the final VLM layer as a

shared latent representation, denoted as $\mathcal{H}_t$,

$$\mathcal{H}_t = \text{VLM}_\theta^{\text{\textcolor{orange}{\char"1F525}}}(\mathcal{I}_t, \mathcal{B}_t, \mathcal{A}_{t-1}, \mathcal{T}_t), \quad (1)$$

where $\mathcal{H}_t$ serves as the common feature space for both future imagination and future action prediction. During this stage, *DriveWorld-VLA* is trained to jointly perform **future imagination** and **action prediction** based on shared latent representation, facilitating the transfer of world model knowledge into the VLA.

**Future imagination** is modeled in the BEV space. Denoiser takes $\mathcal{H}_t$ and $\mathcal{B}_t$ as input to predict future BEV states $\mathcal{B}_{t+\Delta t}$ as below,

$$\begin{aligned} \mathcal{B}_t' &= \text{CROSSATTN}_\theta^{\text{\textcolor{orange}{\char"1F525}}}(\mathcal{B}_t, \mathcal{H}_t, \mathcal{H}_t), \\ \mathcal{B}_{t+\Delta t} &= \text{DENOISER}_\theta^1(\mathcal{H}_t, \mathcal{B}_t', \mathcal{A}_{t-1}), \end{aligned} \quad (2)$$

which are then decoded by a lightweight segmentation head SEG,

$$\mathcal{S}_{t+\Delta t} = \text{SEG}_\theta(\mathcal{B}_{t+\Delta t}), \mathcal{S}_t = \text{SEG}_\theta(\mathcal{B}_t'). \quad (3)$$

DENOISER comprises a history-conditioned branch and a future action-conditioned branch. At this stage, only history-conditioned branch is activated and enforce reasoning from historical observations. This lightweight branch provides dense future supervision, enabling predictive representation learning and effective training of images to BEV tokenizers. The future action-conditioned branch follows a generative paradigm to model controllable future evolution under different action sequences.

**Action prediction** is formulated as trajectory forecasting. A lightweight action decoder takes $\mathcal{H}_t$, $\mathcal{B}_t$, and $\mathcal{A}_{t-1}$ as input, and outputs the predicted future actions $\mathcal{A}_{t+\Delta t}'$:

$$\mathcal{A}_{t+\Delta t}' = \text{ACT}_\theta(\mathcal{H}_t, \mathcal{B}_t, \mathcal{A}_{t-1}), \quad (4)$$

where ACT denotes an action decoder.

**Supervision.** The future BEV state is supervised by decoding a semantic BEV map, while the action decoder is supervised via imitation learning on expert actions. The overall loss at this stage is defined as:

$$\mathcal{L}_{s_1} = \mathcal{L}_{seg} + \mathcal{L}_{act}, \quad (5)$$

where $\mathcal{L}_{seg}$ supervises the semantic BEV map decoding and $\mathcal{L}_{act}$ supervises the predicted actions.

### 3.2. Action Controllability Fine-Tuning

During co-training, *DriveWorld-VLA* is not conditioned on future actions, which prevents it from imagining outcomes based on prospective actions. As a result, *DriveWorld-VLA* cannot form a closed-loop reasoning between actions and

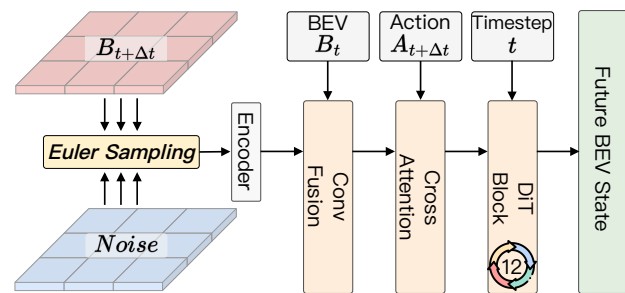

*Figure 3.* **The structure of Action-conditioned Flow-matching Denoiser.** The Denoiser conditions the flow-matching process on the BEV state and GT future actions. The BEV state is processed through LayerNorm, followed by scaling and Embedding. These features are then passed through DiT blocks to perform denoising and generate the future BEV states.

future scene generation. Ideally, *DriveWorld-VLA* should be aware of how its actions affect the future evolution of the environment to evaluate action quality, rather than solely extrapolating from historical observations.

Motivated by this limitation, this stage focuses on endowing *DriveWorld-VLA* with the capability of action-conditioned future imagination. Due to the absence of sensor observations in the BEV space, we adopt an explicit feature-level supervision strategy for BEV prediction, which fundamentally differs from downstream-task supervision used in prior works such as WoTE (Li et al., 2025c) and LAW (Li et al., 2024a). After co-training, $\mathcal{B}_t'$ produced by the BEV tokenizer and VLM can be reliably decoded into a semantic BEV map. We therefore treat this BEV latent space as a pretrained variational representation. Given future multi-view images $\mathcal{I}_{t+\delta t}$, we reuse the Stage 1 encoding pipeline to obtain the corresponding ground truth (GT) BEV latent representation $\mathcal{B}_{t+\Delta t}'$ as follows:

$$\mathcal{H}_{t+\Delta t} = \text{VLM}_\theta^*(\mathcal{I}_{t+\Delta t}, \mathcal{B}_{t+\Delta t}, \mathcal{A}_{t+\Delta t}, \mathcal{T}_{t+\Delta t}), \quad (6)$$

$$\mathcal{B}_{t+\Delta t}' = \text{CROSSATTN}_\theta^*(\mathcal{B}_{t+\Delta t}, \mathcal{H}_{t+\Delta t}, \mathcal{H}_{t+\Delta t}). \quad (7)$$

Subsequently, the second branch of the denoiser employs a DiT-based architecture to learn an action-conditioned flow-matching denoising process, using the BEV state $\mathcal{B}_t'$ and the GT future action $\mathcal{A}_{t+\Delta t}$ as conditions, as shown in Figure 3. The supervision is defined as:

$$\mathcal{L}_{FM} = ||\text{DIT}_\theta(\mathcal{B}_t', \mathcal{A}_{t+\Delta t}, x_k, \frac{k}{N}) - (\mathcal{B}_{t+\Delta t}' - x_0)||^2, \quad (8)$$

where $\text{DIT} = \text{DENOISER}^2$ denotes the second branch of the denoiser and $x_0 \sim \mathcal{N}(\mathbf{0}, \mathbf{I})$, $k$ represents timestep in the flow matching process and is uniformly sampled from the interval [1,N].

**Supervision.** In Action Controllablity Fine-Tuning stage, the overall loss $\mathcal{L}_{s_2}$ consists solely of the loss $\mathcal{L}_{FM}$.

## 3.3. Future-Guided Evaluation & Refinement

This stage aims to establish a closed-loop interaction between action prediction and future imagination based on the highly shared representation $\mathcal{H}_t$ between the VLA and the world model. *DriveWorld-VLA* is required not only to predict actions but also to effectively imagine the corresponding future outcomes induced by those actions, and to evaluate and refine actions according to the imagined future states. Given the $\mathcal{B}_t$, $\mathcal{H}_t$ and $\mathcal{A}_{t-1}$, *DriveWorld-VLA* first predicts future actions $\mathcal{A}'_{t+\Delta t}$ and generates future imagination $\mathcal{B}_{t+\Delta t}$ via the first denoising branch. The predicted actions are subsequently used to condition the second denoising branch, where Euler-based sampling is performed to produce action-conditioned future imagination $\mathcal{B}'_{t+\Delta t}$:

$$\mathcal{B}^{k+1}_{t+\Delta t} = \mathcal{B}^k_{t+\Delta t} + \frac{1}{N}(\text{DiT}_\theta(\mathcal{B}'_t, \mathcal{A}_{t+\Delta t}, x_k, \frac{k}{N})), \quad (9)$$

where $\mathcal{B}'_{t+\Delta t} = \mathcal{B}^{k+1}_{t+\Delta t}$, and $N$ denotes the sampling steps and is set to 25.

To evaluate the quality of predicted actions, we jointly consider the consistency between the action-conditioned imagination $\mathcal{B}'_{t+\Delta t}$ and the corresponding future BEV representation $\mathcal{B}_{t+\Delta t}$. A learned reward function $\mathcal{R}$ assigns a scalar score $\hat{r}_{t+\Delta t}$ to each predicted trajectory, with ground-truth rewards obtained by executing the predicted trajectories in a simulator and performing online evaluation. This process can be formulated as,

$$\hat{r}_{t+\Delta t} = \mathcal{R}(\mathcal{B}'_{t+\Delta t}, \mathcal{B}_{t+\Delta t}, \mathcal{A}'_{t+\Delta t}). \quad (10)$$

Beyond trajectory quality assessment, this reward-driven design promotes close-loop between future imagination and action generation. Rather than uniformly supervising all multi-modal predictions, training prioritizes trajectories with higher predicted rewards, reinforcing those that lead to more favorable imagined outcomes and enabling consequence-aware action refinement,

$$\mathcal{L}'_{act} = \hat{r}_{t+\Delta t} \cdot ||\mathcal{A}'_{t+\Delta t} - \mathcal{A}_{t+\Delta t}||^2. \quad (11)$$

**Supervision.** During this stage, DENOISER and VLM are frozen. Training focuses on refining the reward function and the action head. The future BEV latents generated by the two denoising branches are fused and fed into the segmentation decoder for supervised BEV decoding, resulting in three complementary supervision signals:

$$\mathcal{L}_{s_3} = \mathcal{L}'_{act} + \mathcal{L}_{seg} + \mathcal{L}_{rew}, \quad (12)$$

where $\mathcal{L}_{seg}$ also supervises the semantic BEV map decoding, $\mathcal{L}'_{act}$ supervises the predicted actions weighted by rewards and $\mathcal{L}_{rew}$ supervises the reward function $\mathcal{R}$.

## 4. Experiments

### 4.1. Details

**Dataset and Metrics.** We evaluate *DriveWorld-VLA* on NAVSIMv1 (Dauner et al., 2024), NAVSIMv2 (Cao et al., 2025) and nuScenes (Caesar et al., 2020). **NAVSIMv1** is an autonomous driving planning benchmark built on the Open-Scene (Contributors, 2023) dataset, providing 120 hours of driving data at 2 Hz with multi-camera inputs and fused LiDAR point clouds. NAVSIMv1 adopts a non-reactive open-loop simulation protocol, where the core metric Predictive Driver Model Score (PDMS) is defined as the product of a penalty term and a weighted average score. The penalty term reflects constraint satisfaction such as No Collision (NC) and Drivable Area Compliance (DAC), while the weighted average combines Ego Progress (EP), Time-To-Collision (TTC), and Comfort (C) with weights of 5:5:2. **NAVSIMv2** proposes a two-stage pseudo-simulation evaluation. Its core metric is called the Extended Predictive Driver Model Score (EPDMS), which is constructed from the following metrics: No at-fault Collision (NC), Drivable Area Compliance (DAC), Driving Direction Compliance (DDC), Traffic Light Compliance (TLC), Ego Progress (EP), Time to Collision (TTC), Lane Keeping (LK), History Comfort (HC), and Extended Comfort (EC). For open-loop planning, **nuScenes** is a large-scale outdoor driving dataset with 1000 multi-modal scenes. Each scene lasts 20 s, is annotated at 2 Hz, and includes six synchronized camera images and LiDAR point clouds. We use L2 and Collision Rate (CR) as evaluation metrics. See Appendix A for more details.

**Implementation.** For **NAVSIM**, we concatenate the left-front view, front view, and right-front view in the order to form a 256×1024 composite image as the model input. We use ResNet-34 (He et al., 2016) as the BEV encoder. Training is performed with the AdamW optimizer using an initial learning rate of 1e-4 and a batch size of 16. Each stage is trained for 20 epochs on 8 NVIDIA H20 GPUs, with a total training time of approximately 120 hours. For the **nuScenes** open-loop evaluation, the 6-view input images are resized to 640×384. We adopt a Swin-T (Liu et al., 2021) backbone initialized with pretrained weights and follow BEV-Planner (Li et al., 2024d) to encode the BEV feature map. We train with AdamW using an initial learning rate of 7e-5 and a batch size of 1. Each stage is trained for 24 epochs on 8 NVIDIA H20 GPUs, for a total training time of approximately 93 hours. Note that, for a fair comparison, none of the nuScenes-based experiments uses ego-state information.

### 4.2. Main Results

**NAVSIMv1.** Table 1 reports the closed-loop planning performance on NAVSIMv1. Our *DriveWorld-VLA* achieves a PDMS of 91.3, outperforming top methods across dif-

*Table 1.* Comparison with state-of-the-art methods on the **NAVSIMv1** (Dauner et al., 2024). Abbreviations: 1×C (front single-view camera), N×C (surround multi-view cameras), L (LiDAR), NC (No Collision), DAC (Drivable Area Compliance), TTC (Time-To-Collision), EP (Ego Process), C (Comfort), PDMS (Predictive Driver Model Score).

| Methods | Venue | Sensors | NC↑ | DAC↑ | TTC↑ | C↑ | EP↑ | PDMS↑ |
|---|---|---|---|---|---|---|---|---|
| Human | - | - | 100.0 | 100.0 | 100.0 | 99.9 | 87.5 | 94.8 |
| *E2E-based Methods* | | | | | | | | |
| TransFuser (Chitta et al., 2022) | TPAMI 2023 | 3×C + L | 97.7 | 92.8 | 92.8 | **100.0** | 79.2 | 84.0 |
| PARA-Drive (Weng et al., 2024) | CVPR 2024 | 6×C | 97.9 | 92.4 | 93.0 | 99.8 | 79.3 | 84.0 |
| Hydra-MDP (Li et al., 2024b) | arXiv 2024 | 3×C + L | 98.3 | 96.0 | 94.6 | **100.0** | 78.7 | 86.5 |
| DiffusionDrive (Liao et al., 2025) | CVPR 2025 | 3×C + L | 98.2 | 96.2 | 94.7 | **100.0** | 82.2 | 88.1 |
| *World-Model-based Methods* | | | | | | | | |
| LAW (Li et al., 2024a) | ICLR 2024 | 1×C | 96.4 | 95.4 | 88.7 | 99.9 | 81.7 | 84.6 |
| DrivingGPT (Chen et al., 2025) | ICCV 2025 | 1×C | 98.9 | 90.7 | 94.9 | 95.6 | 79.7 | 82.4 |
| WoTE (Li et al., 2025c) | ICCV 2025 | 3×C + L | 98.5 | 96.8 | 94.4 | 99.9 | 81.9 | 88.3 |
| Epona (Zhang et al., 2025) | ICCV 2025 | 3×C | 97.9 | 95.1 | 93.8 | 99.9 | 80.4 | 86.2 |
| *VLA-based Methods* | | | | | | | | |
| AutoVLA (Zhou et al., 2025c) | NeurIPS 2025 | 3×C | 98.4 | 95.6 | **98.0** | 99.9 | 81.9 | 89.1 |
| ReCogDrive (Li et al., 2025d) | ICLR 2026 | 3×C | 98.2 | 97.8 | 95.2 | 99.8 | 83.5 | 89.6 |
| DriveVLA-W0 (Li et al., 2025b) | ICLR 2026 | 1×C | 98.7 | **99.1** | 95.3 | 99.3 | 83.3 | 90.2 |
| **DriveWorld-VLA (Ours)** | - | 3×C | **99.1** | 98.2 | 96.1 | **100.0** | **85.9** | **91.3** |

*Table 2.* Comparison with state-of-the-art methods on the **NAVSIMv2** (Cao et al., 2025). Abbreviations: NC (No at-fault Collision), DAC (Drivable Area Compliance), DDC (Driving Direction Compliance), TLC (Traffic Light Compliance), EP (Ego Progress), TTC (Time to Collision), LK (Lane Keeping), HC (History Comfort), EC (Extended Comfort), EPDMS (Extended Predictive Driver Model Score).

| Methods | NC↑ | DAC↑ | DDC↑ | TLC↑ | EP↑ | TTC↑ | LK↑ | HC↑ | EC↑ | EPDMS↑ |
|---|---|---|---|---|---|---|---|---|---|---|
| *E2E-based Methods* | | | | | | | | | | |
| TransFuser (Chitta et al., 2022) | 96.9 | 89.9 | 97.8 | 99.7 | 87.1 | 95.4 | 92.7 | **98.3** | 87.2 | 76.7 |
| DiffusionDrive (Liao et al., 2025) | 98.2 | 95.9 | 99.4 | **99.8** | 87.5 | 97.3 | 96.8 | **98.3** | **87.7** | 84.5 |
| HydraMDP++ (Li et al., 2025a) | 97.2 | 97.5 | 99.4 | 99.6 | 83.1 | 96.5 | 94.4 | 98.2 | 70.9 | 81.4 |
| DriveSuprim (Yao et al., 2025) | 97.5 | 96.5 | 99.4 | 99.6 | **88.4** | 96.6 | 95.5 | **98.3** | 77.0 | 83.1 |
| ARTEMIS (Feng et al., 2025a) | 98.3 | 95.1 | 98.6 | **99.8** | 81.5 | 97.4 | 96.5 | **98.3** | - | 83.1 |
| *VLA-based Methods* | | | | | | | | | | |
| DriveVLA-W0 (Li et al., 2025b) | 98.5 | **99.1** | 98.0 | 99.7 | 86.4 | **98.1** | 93.2 | 97.9 | 58.9 | 86.1 |
| **DriveWorld-VLA (Ours)** | **98.6** | **99.1** | **99.6** | **99.8** | 87.4 | 97.9 | **97.0** | 97.8 | 78.6 | **86.8** |

ferent paradigms, including DiffusionDrive (Liao et al., 2025), WoTE (Li et al., 2025c), and DriveVLA-W0 (Li et al., 2025b). Notably, *DriveWorld-VLA* attains an NC of 99.1 and an EP of 85.9, suggesting that our policy effectively enforces safety constraints while sustaining efficient forward progress.

**NAVSIMv2.** Table 2 reports the closed-loop planning performance on NAVSIMv2. *DriveWorld-VLA* achieves an EPDMS of 86.8, again outperforming all compared methods. Particularly, it performs excellently in DAC, DDC, and LK, achieving 99.1, 99.6, and 97.0, respectively.

**nuScenes.** Table 3 reports the 3-second planning results on the nuScenes (Caesar et al., 2020) validation dataset. The performance of recent methods suggests that short-horizon planning has been extensively studied. Compared with E2E-based methods and world-model-based methods, our *DriveWorld-VLA* demonstrates a clear advantage, achieving an average L2 of 0.61m and a collision rate as low as 0.16%. Even relative to the state-of-the-art approaches FSDrive (Zeng et al., 2025) and HERMES-p (Zhou et al., 2025b), *DriveWorld-VLA* still maintains a decisive advantage in terms of collision rate. These results indicate that

the policy of *DriveWorld-VLA* is beneficial throughout the entire short-horizon planning process. We reiterate that ego-state information is disabled.

### 4.3. Ablation Study

**Ablation on Training Process.** Table 4 presents the performance of different stages of the *DriveWorld-VLA* training process across various datasets. For NAVSIMv1 (Dauner et al., 2024), each training stage significantly improves the model's performance, with PDMS achieving increases of +1.9 and +1.8, respectively. For nuScenes (Caesar et al., 2020), the positive impact of Stage 2 is more pronounced. Short-term planning in the 2nd and 3rd seconds shows almost no improvement from Stage 3, with only -0.01 and -0.02 changes. We propose two hypotheses to explain this phenomenon. First, open-loop planning tasks do not necessarily benefit from generative supervision at all times. Second, the Reward Model has a greater effect on closed-loop planning than on open-loop planning. From an engineering perspective, *DriveWorld-VLA* uses different baselines (Li et al., 2025c; 2024a) and Reward Models across the two benchmarks. From a theoretical standpoint, the integration

*Table 3.* Comparison with state-of-the-art methods on the **nuScenes** (Caesar et al., 2020) validation dataset. * denotes results reproduced with the official checkpoint. Abbreviations: C (surround multi-view cameras), CR (Collision Rate).

| Methods | Venue | Sensors | L2(m)↓ | | | | CR(%)↓ | | | |
|---|---|---|---|---|---|---|---|---|---|---|
| | | | 1s | 2s | 3s | Avg. | 1s | 2s | 3s | Avg. |
| *E2E-based Methods* | | | | | | | | | | |
| UniAD (Hu et al., 2023) | CVPR 2023 | C | 0.48 | 0.96 | 1.65 | 1.03 | 0.05 | 0.17 | 0.71 | 0.31 |
| VAD (Jiang et al., 2023) | ICCV 2023 | C | 0.41 | 0.70 | 1.05 | 0.72 | 0.07 | 0.17 | 0.41 | 0.22 |
| MomAD (Song et al., 2025a) | CVPR 2025 | C | 0.43 | 0.88 | 1.62 | 0.98 | 0.06 | 0.16 | 0.68 | 0.30 |
| *World-Model-based Methods* | | | | | | | | | | |
| LAW* (Li et al., 2024a) | ICLR 2024 | C | 0.31 | 0.61 | 1.02 | 0.65 | 0.27 | 0.21 | 0.54 | 0.34 |
| GenAD (Zheng et al., 2024) | ECCV 2024 | C | 0.36 | 0.83 | 1.55 | 0.91 | 0.06 | 0.23 | 1.00 | 0.43 |
| Epona (Zhang et al., 2025) | ICCV 2025 | C | 0.61 | 1.17 | 1.98 | 1.25 | 0.01 | 0.22 | 0.85 | 0.36 |
| FSDrive (Zeng et al., 2025) | NeurIPS 2025 | C | 0.28 | 0.52 | 0.80 | 0.53 | 0.06 | 0.13 | **0.32** | 0.17 |
| *VLA-based Methods* | | | | | | | | | | |
| HERMES-p (Zhou et al., 2025b) | ICCV 2025 | C | **0.16** | **0.32** | **0.59** | **0.36** | **0.00** | 0.14 | 0.82 | 0.32 |
| **DriveWorld-VLA (Ours)** | - | C | 0.28 | 0.58 | 0.99 | 0.61 | **0.00** | **0.10** | 0.38 | **0.16** |

*Table 4.* Ablation study of **Training Process** on **NAVSIMv1** (Dauner et al., 2024) and **nuScenes** (Caesar et al., 2020) validation dataset.

| Training Process | | | NAVSIMv1 (Closed-Loop) | | | | | | nuScenes (Open-Loop) | | | |
|---|---|---|---|---|---|---|---|---|---|---|---|---|
| Stage 1 | Stage 2 | Stage 3 | NC↑ | DAC↑ | TTC↑ | C↑ | EP↑ | PDMS↑ | CR↓@1s | CR↓@2s | CR↓@3s | CR↓@Avg. |
| ✗ | ✗ | ✗ | 98.5 | 95.8 | 94.4 | 99.9 | 80.9 | 87.1 | 0.27 | 0.21 | 0.54 | 0.34 |
| ✓ | ✗ | ✗ | 98.6 | 96.1 | 95.1 | 99.9 | 81.0 | 87.6 | 0.09 | 0.19 | 0.48 | 0.25 |
| ✓ | ✓ | ✗ | 98.9 | 97.3 | 94.4 | 100.0 | 84.5 | 89.5 | 0.05 | 0.11 | 0.40 | 0.19 |
| ✓ | ✓ | ✓ | 99.1 | 98.2 | 96.1 | 100.0 | 85.9 | 91.3 | 0.00 | 0.10 | 0.38 | 0.16 |

*Table 5.* Ablation study of **Pipeline Strategies** on **NAVSIMv1** (Dauner et al., 2024). 'Non-progressive 1' denotes that Stage 2 and Stage 3 are trained simultaneously after the completion of Stage 1, 'Non-progressive 2' denotes that Stage 1 and Stage 2 are trained simultaneously before Stage 3, while 'Progressive' denotes the strategy adopted in our work.

| Pipeline Strategies | NC↑ | DAC↑ | TTC↑ | C↑ | EP↑ | PDMS↑ |
|---|---|---|---|---|---|---|
| Non-progressive 1 | 97.8 | 94.5 | 93.5 | 99.9 | 75.6 | 83.6 |
| Non-progressive 2 | 84.2 | 74.3 | 74.9 | 100.0 | 45.7 | 52.5 |
| Progressive | 99.1 | 98.2 | 96.1 | 100.0 | 85.9 | 91.3 |

*Table 6.* Ablation study of **VLM Strategies** on **NAVSIMv1** (Dauner et al., 2024). 'Freeze' denotes whether the parameters of VLM are also frozen during Stage 1. 'Pre-train' denotes whether the pre-training strategy aligned with ReCogDrive (Li et al., 2025d) is adopted.

| VLM Strategies | | NC↑ | DAC↑ | TTC↑ | C↑ | EP↑ | PDMS↑ |
|---|---|---|---|---|---|---|---|
| Freeze | Pre-train | | | | | | |
| ✓ | ✗ | 98.4 | 95.7 | 96.2 | 99.9 | 80.4 | 87.2 |
| ✓ | ✓ | 98.6 | 96.1 | 95.1 | 99.9 | 81.0 | 87.6 |
| ✗ | ✓ | 98.7 | 96.7 | 95.6 | 99.9 | 83.1 | 88.3 |

of generative tasks and planning tasks is constrained by the feedback mechanism within the inference process.

**Ablation on Pipeline Strategies.** Table 5 shows the performance of *DriveWorld-VLA* under different pipeline strategies. The 'Non-progressive 1' strategy refers to updating the parameters of both the future imagination branch and the action prediction branch after Stage 1. The 'Non-progressive 2' strategy refers to training Stage 1 and the future imagination branch simultaneously before the action

prediction branch. The 'Progressive' strategy follows the training scheme outlined in Figure 2. For a fair comparison, both non-progressive strategies enable the reward model and adopt double the training length. Clearly, both 'Non-progressive' strategies result in a significant performance drop, with 'Non-progressive 1' achieving -7.7 PDMS and 'Non-progressive 2' suffering a much more severe drop of -38.8 PDMS, which indicates that our strategy is effective and non-redundant. The model must first learn from the GT action and enhance the latent feature space before benefiting from action prediction. Additionally, 'Non-progressive 2' makes the generative model hard to converge, forcing Stage 3 to optimize on noisy signals and leading to severe performance degradation. This further demonstrates that even within the same feature space, generative and planning tasks need to be asynchronously unified.

**Ablation on VLM Strategies.** Table 6 shows the different VLM strategies used during the training of *DriveWorld-VLA*. 'Freeze' indicates whether the VLM parameters are frozen during Stage 1, and 'Pre-Train' indicates whether the VLM is pre-trained and fine-tuned following the settings in ReCogDrive (Li et al., 2025d) (conducting 3 epochs of SFT using dataset constructed by the hierarchical data pipeline). The optimal strategy is shown in the third row of the table. Both omitting pre-training and fully freezing the VLM parameters limit the model's performance. This indicates that while non-targeted pre-training can contribute, it does not fully unlock the VLM's potential. The VLM parameters must undergo optimization through a learning process during the initial stage to accurately model shared space

*Table 7.* Ablation study of **Supervision** on **NAVSIMv1** (Dauner et al., 2024). 'Task' denotes task-level supervision and 'Features' denotes feature-level supervision.

| Supervision | | NC↑ | DAC↑ | TTC↑ | C↑ | EP↑ | PDMS↑ |
|---|---|---|---|---|---|---|---|
| Task | Features | | | | | | |
| ✓ | ✗ | 98.7 | 96.3 | 94.0 | 100.0 | 82.9 | 87.9 |
| ✓ | ✓ | 99.1 | 98.2 | 96.1 | 100.0 | 85.9 | 91.3 |

*Table 8.* Ablation study of **VLM Perturbation** on **NAVSIMv1** (Dauner et al., 2024). The perturbation is drawn from a Gaussian distribution $\mathcal{N}(0, 0.05)$.

| VLM Perturbation | NC↑ | DAC↑ | TTC↑ | C↑ | EP↑ | PDMS↑ |
|---|---|---|---|---|---|---|
| Stage 1 | 46.9 | 41.8 | 36.7 | 12.1 | 9.3 | 7.3 |
| Stage 3 | 46.2 | 40.0 | 36.0 | 12.2 | 8.8 | 6.9 |
| None | 99.1 | 98.2 | 96.1 | 100.0 | 85.9 | 91.3 |

*Table 9.* Ablation study of **Text Masking** on **NAVSIMv1** (Dauner et al., 2024).

| Text Masking | NC↑ | DAC↑ | TTC↑ | C↑ | EP↑ | PDMS↑ |
|---|---|---|---|---|---|---|
| ✓ | 98.2 | 94.0 | 94.4 | 99.9 | 75.6 | 84.1 |
| ✗ | 99.1 | 98.2 | 96.1 | 100.0 | 85.9 | 91.3 |

*Table 10.* **Inference Speed**. All results are validated on a single NVIDIA H20 GPU.

| Methods | Function | | FPS(Hz) |
|---|---|---|---|
| | Generation | Planning | |
| Vista(Gao et al., 2024b) | ✓ | ✗ | 0.57 |
| Epona(Zhang et al., 2025) | ✓ | ✓ | 0.40 |
| **DriveWorld-VLA (Ours)** | ✓ | ✓ | **1.16** |

features and enhance the model's performance. Considering the future need for lightweight design, we believe that completely abandoning VLM updates is also acceptable.

**Ablation on Supervision.** Table 7 shows the impact of task-level supervision and feature-level supervision on *DriveWorld-VLA*. $\mathcal{L}seg$ and $\mathcal{L}act$ are considered task-level supervision, while the supervision from the denoising process is considered feature-level supervision. We introduce noise that follows the distribution $\mathcal{N}(0, 5)$ into the latent variables during the inference phase, in order to mitigate the constraining effect of feature-level supervision. The results show that weakening feature-level supervision leads to a decline in model performance. Relying solely on task-level supervision causes the model to lack fine-grained feature guidance, which in turn affects performance.

**Ablation on VLM Perturbation.** To verify the effectiveness of the shared latent space built by *DriveWorld-VLA*, we conduct perturbation experiments on VLM latent states, as shown in Table 8. Perturbing VLM latent states causes significant performance drops in Stage 1 and Stage 3, even though Stage 3 receives no direct supervision on latent states.

This indicates that the model cannot maintain downstream task performance without relying on shared features, which are effectively exploited for action generation and future scenario imagination.

**Ablation on Text Masking.** Table 9 presents the results under different text mask settings. During inference, we replace textual inputs with empty strings to simulate scenarios without language instructions. The planning performance drops remarkably after text masking. In *DriveWorld-VLA*, textual prompts do not participate in decision-making via step-by-step reasoning chains. Nevertheless, this does not mean the model merely treats text as trivial auxiliary context. In our design, textual prompts serve as conditional constraints to directly regulate action generation.

### 4.4. Inference Speed

Table 10 reports the inference speed of different methods, all measured on a single NVIDIA H20 GPU for fair comparison. Even though *DriveWorld-VLA* achieves the highest FPS, efficiency challenges are common among methods with similar technical pipelines (Jia et al., 2025; Mousakhan et al., 2025). As such, this does not imply that *DriveWorld-VLA* holds a dedicated advantage in raw inference speed. The core focus of *DriveWorld-VLA* is to improve planning performance, and efficiency optimization is left for future.

### 4.5. Robustness Study

Table 11 presents the robustness evaluation results based on two corrupted benchmark datasets, namely NAVSIM-C and nuScenes-C. nuScenes-C (Dong et al., 2023) is a corruption benchmark derived from nuScenes (Caesar et al., 2020), which contains 27 corruption types with five severity levels. We focus on four representative corruption types, including adverse weather corruptions (Snow, Rain, and Fog) and sensor/motion corruption (Motion Blur). NAVSIM-C (from NAVSIMv1 (Dauner et al., 2024)) adopts the same configuration as nuScenes-C, with the corruption severity fixed at level 5. Results demonstrate that *DriveWorld-VLA* achieves the optimal robustness across all corruption types on both robustness benchmarks. On the NAVSIM-C benchmark, Motion Blur (-18.3) and Fog (-14.5) cause dramatic performance degradation. By contrast, on the nuScenes-C benchmark, Rain (+0.37/+0.24) and Snow (+0.19/+0.11) bring relatively mild performance variations.

### 4.6. Visualization

As shown in Figure 4, we provide some visual examples stemming from NAVSIM (Dauner et al., 2024) for *DriveWorld-VLA*. By the Figure, *DriveWorld-VLA* with three-stage training scheme empowered by future imagination demonstrates excellent and progressive planning results.

*Table 11.* Robustness study of planning performance on **NAVSIM-C** and **nuScenes-C**. 'Clean' denotes the dataset without artificial corruption, and four typical 'Corruptions' (Dong et al., 2023) including 'Snow', 'Rain', 'Fog', and 'Motion Blur' are used to simulate challenging real-world scenarios.

| Benchmark | Method | Clean | Corruptions | | | |
|---|---|---|---|---|---|---|
| | | | Motion Blur | Snow | Rain | Fog |
| NAVSIM-C (PDMS↑) | TransFuser(Chitta et al., 2022) | 84.0 | 65.8-18.2 | 72.3-11.7 | 69.2-14.8 | 73.0-11.0 |
| | **DriveWorld-VLA (Ours)** | **91.3** | **73.0**-18.3 | **80.6**-10.7 | **85.2**-6.1 | **76.8**-14.5 |
| nuScenes-C (L2↓ / CR↓@Avg.) | LAW(Li et al., 2024a) | 0.65/0.34 | 0.74/0.56+0.09/+0.22 | 0.92/0.44+0.27/+0.10 | 1.00/0.54+0.35/+0.20 | 0.68/0.55+0.03/+0.21 |
| | **DriveWorld-VLA (Ours)** | **0.61/0.16** | **0.62/0.45**+0.01/+0.29 | **0.80/0.27**+0.19/+0.11 | **0.98/0.40**+0.37/+0.24 | **0.66/0.20**+0.05/+0.04 |

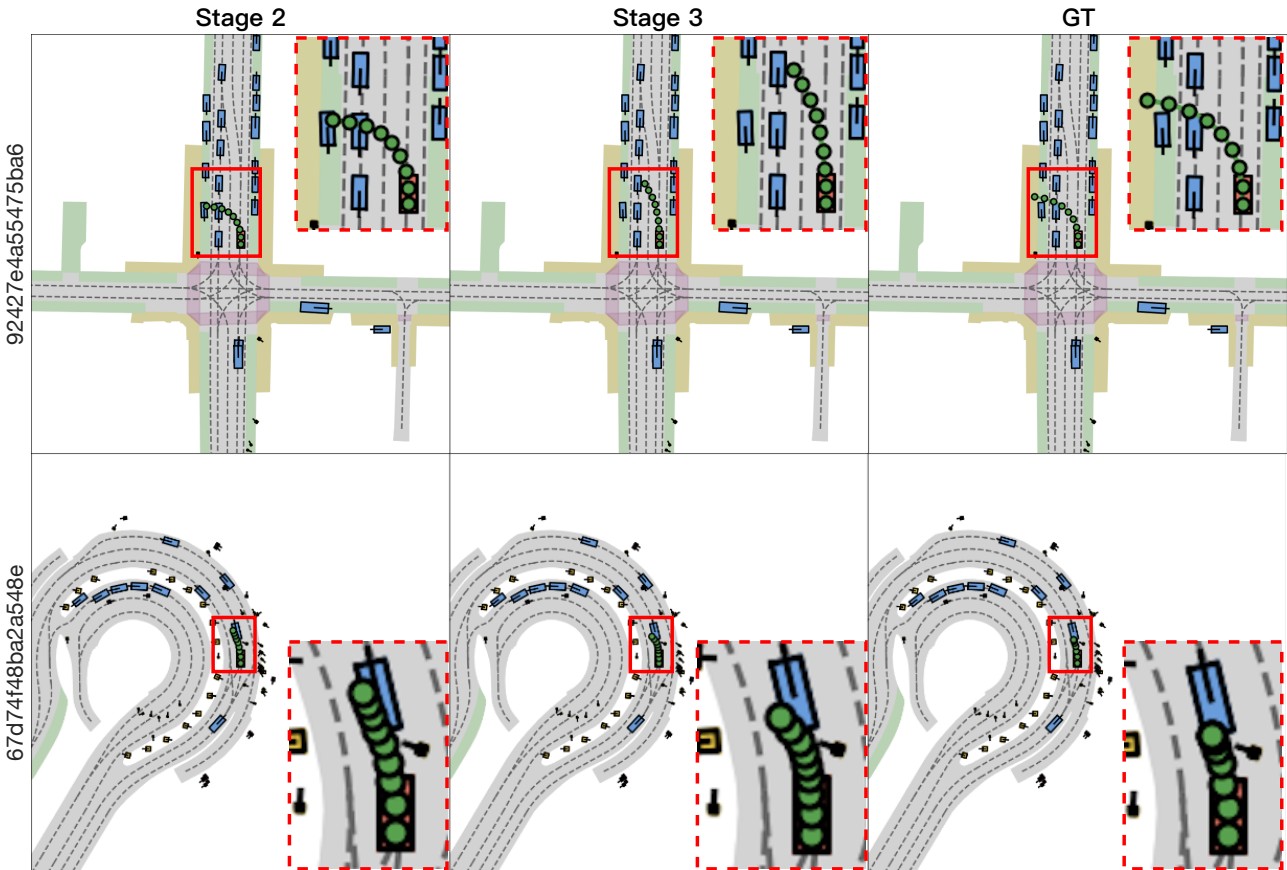

*Figure 4.* **The 4s trajectory planning visualization examples from NAVSIM (Dauner et al., 2024) for DriveWorld-VLA.** Stage 2 generates predictions similar to the GTs, but with a higher collision risk. In contrast, Stage 3 introduces future imagination, resulting in more robust predictions and significantly reducing the collision risk. The changes observed across the training stages confirm the accuracy of world modeling and its understanding of physical dynamics. Left label: sample tokens. Top label: source of trajectory.

See Appendix B for more illustration samples (Figures S2-S6 for NAVSIM and Figures S7-S8 for nuScenes).

## 5. Conclusion

In this study, we introduce ***DriveWorld-VLA***, a novel framework designed to enhance autonomous driving decision-making and forward-looking reasoning by tightly integrating VLA and world models. *DriveWorld-VLA* shares scene representations within the latent space, allowing the VLA to benefit more effectively from the world model. This integration enables direct access to global information regarding the future evolution of scenes, which is then applied to the decision-making process. Unlike existing approaches, *DriveWorld-VLA* uses the latent states of the world model as the basis for decision-making, assisting the system in evaluating the long-term impact of actions on future scenarios. We conduct extensive closed-loop and open-loop planning evaluations of *DriveWorld-VLA* across multiple benchmarks. The results demonstrate that *DriveWorld-VLA* significantly outperforms current state-of-the-art methods, showcasing its immense potential in decision-making.

## Acknowledgements

This work is accomplished under the support of Xiaomi EV. This work is supported by the National Natural Science Foundation of China (NSFC) under Grants No. 62536001 (Key Program) and No. 62576026.

## Impact Statement

This paper presents work aimed at advancing the field of end-to-end autonomous driving by integrating Vision-Language-Action (VLA) and World Models to enhance decision-making and forward-looking imagination. The datasets and references used in this work are publicly available, and the goal is to improve the safety of autonomous driving systems. We believe this work has positive societal implications, particularly in the context of autonomous vehicle technology, by contributing to safer transportation systems. Additionally, there are no specific ethical concerns that we feel must be highlighted here at this stage.

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

# A. More Experiment Details.

### A.1. Prompt

In the system prompt for InternVL (Zhu et al., 2025), we include a description of the task, metrics, input and output. Figure S1 is an example for the two benchmarks (Dauner et al., 2024; Cao et al., 2025; Caesar et al., 2020).

### A.2. Image Tokens

We explicitly inject visual information into the input sequence in the form of a 'text-domain visual placeholder token sequence'. Given an input image (after view stitching), we first perform adaptive tiling to accommodate varying aspect ratios. Where the image is partitioned into $N_P$ feature patches of size $448 \times 448$. When multiple patches are used, an additional $448 \times 448$ thumbnail is appended to provide a global view. We then insert the textual prompts into a placeholder span delimited by special boundary markers (e.g., `...</img>`), within which a dedicated placeholder token `<IMG_CONTEXT>` is repeated. We assign a fixed number of $K = 256$ `<IMG_CONTEXT>` tokens to each patch. This design ensures that variations in image resolution or the number of patches are directly reflected in the number of text-side image tokens, while the overall sequence is subsequently tokenized and padded to a fixed maximum length.

### A.3. Vision-language Representation

On the vision side, we stack the normalized patches into a tensor in $\mathbb{R}^{P \times 3 \times 448 \times 448}$, together with an accompanying mask that indicates which patches are valid. During multi-modal fusion, the model first locates all occurrences of `<IMG_CONTEXT>` in the text indices, and then applies a visual encoder to extract per-patch features and project them into the language hidden space. The resulting visual features and language tokens are jointly processed by the backbone network. This yields fused sequence hidden states $\mathbf{H} \in \mathbb{R}^{B \times L \times D}$, where $L$ denotes the maximum tokenized sequence length and $D$ is the language backbone hidden dimension. The representation thus captures both textual context and visual information carried at the `<IMG_CONTEXT>` positions. To obtain a fixed-length compact representation, we first linearly project the fused hidden states from $D = 1536$ to a lower dimension $d = 256$, producing $\tilde{\mathbf{H}} \in \mathbb{R}^{B \times L \times d}$. We then introduce a set of learnable latent query vectors $\mathbf{Z}_0 \in \mathbb{R}^{B \times N_L \times d}$, where we set the number of latents to $N_L = 700$. Using cross-attention with $\mathbf{Z}_0$ as queries and $\tilde{\mathbf{H}}$ as keys/values, we aggregate information from a variable-length sequence into a fixed-length representation $\mathbf{Z} \in \mathbb{R}^{B \times N_L \times d}$, which serves as a compact vision–language representation for downstream tasks.

# B. More Visualization.

We have provided more visual examples showed in Figure S2, Figure S3, Figure S4, Figure S5 and Figure S6 for NAVSIM (Dauner et al., 2024), as well as Figure S7, Figure S8 for nuScenes (Caesar et al., 2020).

**For NAVSIM**

You are a vehicle trajectory prediction model for autonomous driving. Your task is to predict the ego vehicle's 4-second trajectory based on the following inputs: multi-view images from 8 cameras, ego vehicle states (position), and discrete navigation commands. The input provides a 2-second history, and your output should ensure a safe trajectory for the next 4 seconds. Your predictions must adhere to the following metrics:

1. Non Collisions (NC): Avoid collisions with other objects/vehicles.
2. Drivable Area Compliance (DAC): Stay within the drivable area.
3. Time to Collision (TTC): Maintain a safe distance from other vehicles.
4. Ego Progress (EP): Ensure the ego vehicle moves forward without being stuck.
5. Comfort (C): Avoid sharp turns and sudden decelerations.
6. Driving Direction Compliance (DDC): Align with the intended driving direction.

For evaluation, use the PDM Score, which combines these metrics: PDM Score = NC * DAC * (5*TTC + 5*EP + 2*C + 0*DDC) / 12.

Your predictions will be evaluated through a non-reactive 4-second simulation with an LQR controller and background actors following their recorded trajectories. The better your predictions, the higher your score.

**For nuScenes**

You are an autonomous vehicle planning model for autonomous driving. Your task is to plan the ego vehicle's 3-second trajectory based on the following inputs: multi-view images from 6 cameras, ego vehicle states (position, velocity), and discrete navigation commands. The input provides a 2-second history, and your output should ensure a safe and efficient trajectory for the next 3 seconds. Your predictions will be evaluated using the following metrics:

1. L2 Distance (L2): The square root of the sum of squared differences between the corresponding coordinates of predicted and actual points, measuring the accuracy of the predicted trajectory.
2. Collision Rate (CR): Measures the safety of the predicted trajectory by quantifying the frequency of collisions with vehicles, pedestrians, and obstacles. The goal is to avoid such collisions.

**Common**

As an autonomous driving system, predict the vehicle's trajectory based on:
1. Visual perception from camera
2. Historical motion context (last 4 timesteps): {history_str}
3. Active navigation command: [{command_str}]
Output requirements: Predict 8 future trajectory points
- Each point format: (x:float, y:float, heading:float)
- Use [PT, ...] to encapsulate the trajectory
- Maintain numerical precision to 2 decimal places

*Figure S1.* **InternVL system prompts.** {history_str} denotes the ground-truth historical trajectory sequence, where each frame in the sequence is represented by a 2D coordinate. {command_str} denotes navigation commands in text form, e.g., 'turn left', 'go straight' or 'turn right'. The full prompt fed into the model is composed of each benchmark's specific prompt and the common prompt.

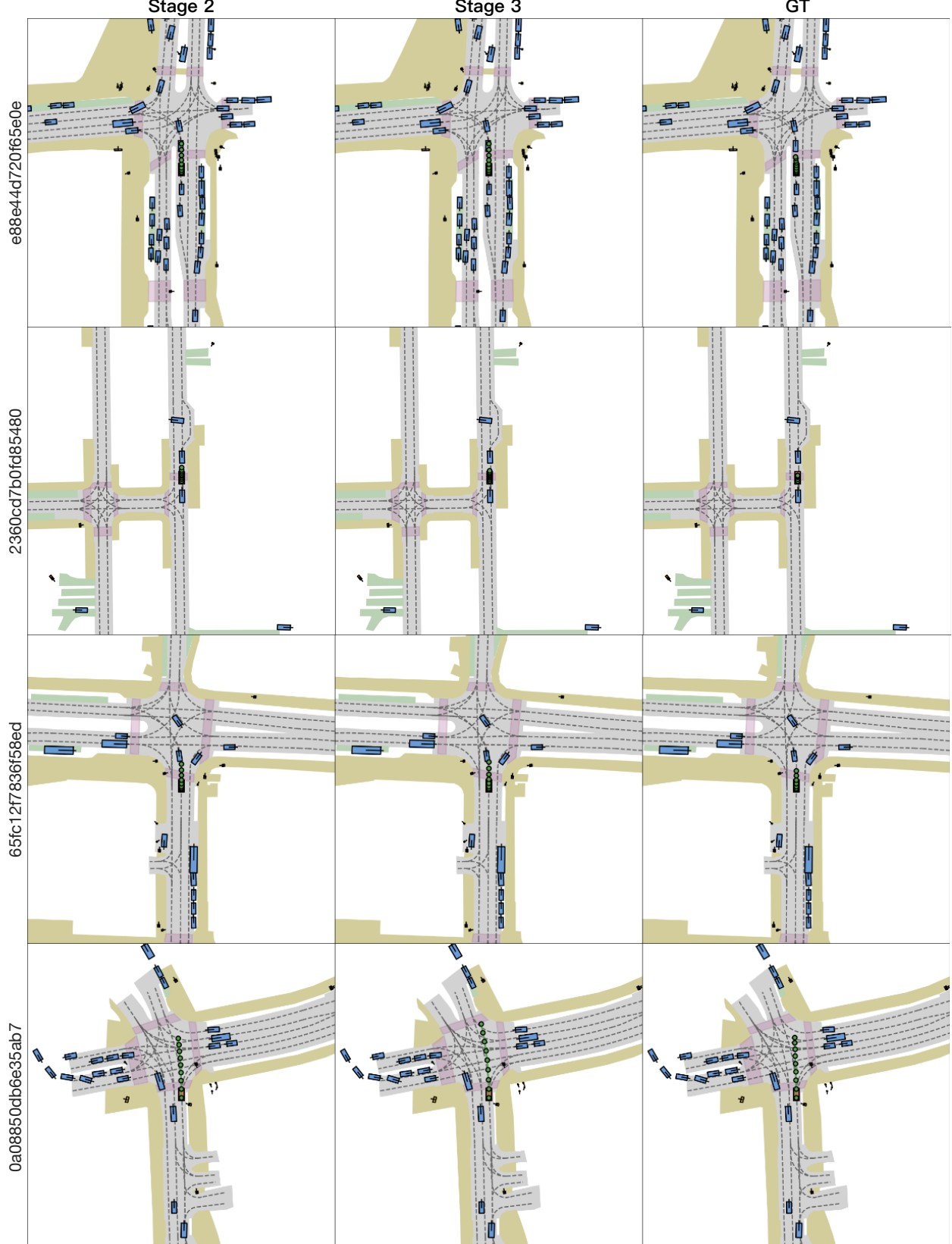

*Figure S2.* Visualization examples of NAVSIM (Dauner et al., 2024). Left label: sample tokens. Top label: source of trajectory.

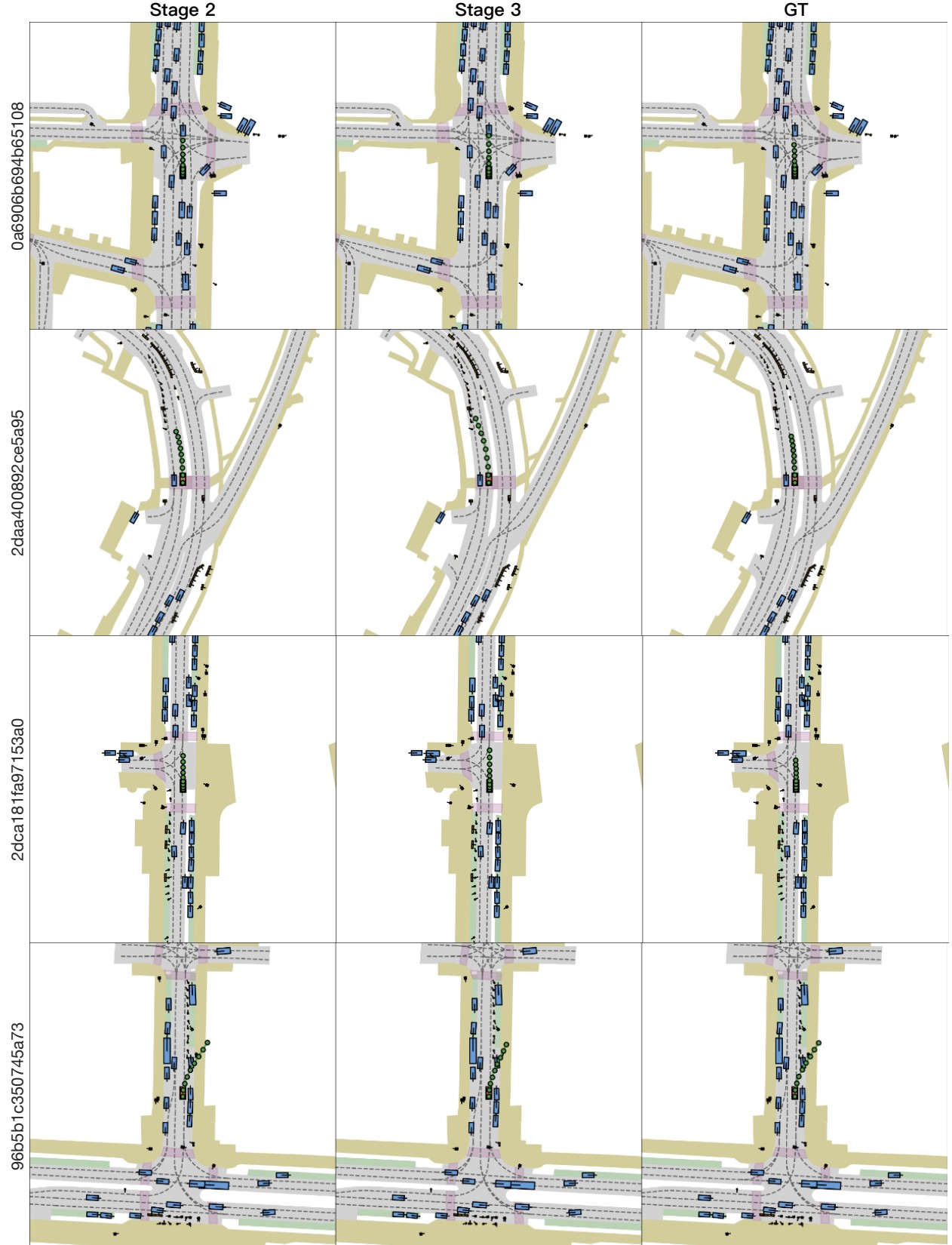

*Figure S3.* Visualization examples (cont.) of NAVSIM (Dauner et al., 2024). Left label: sample tokens. Top label: source of trajectory.

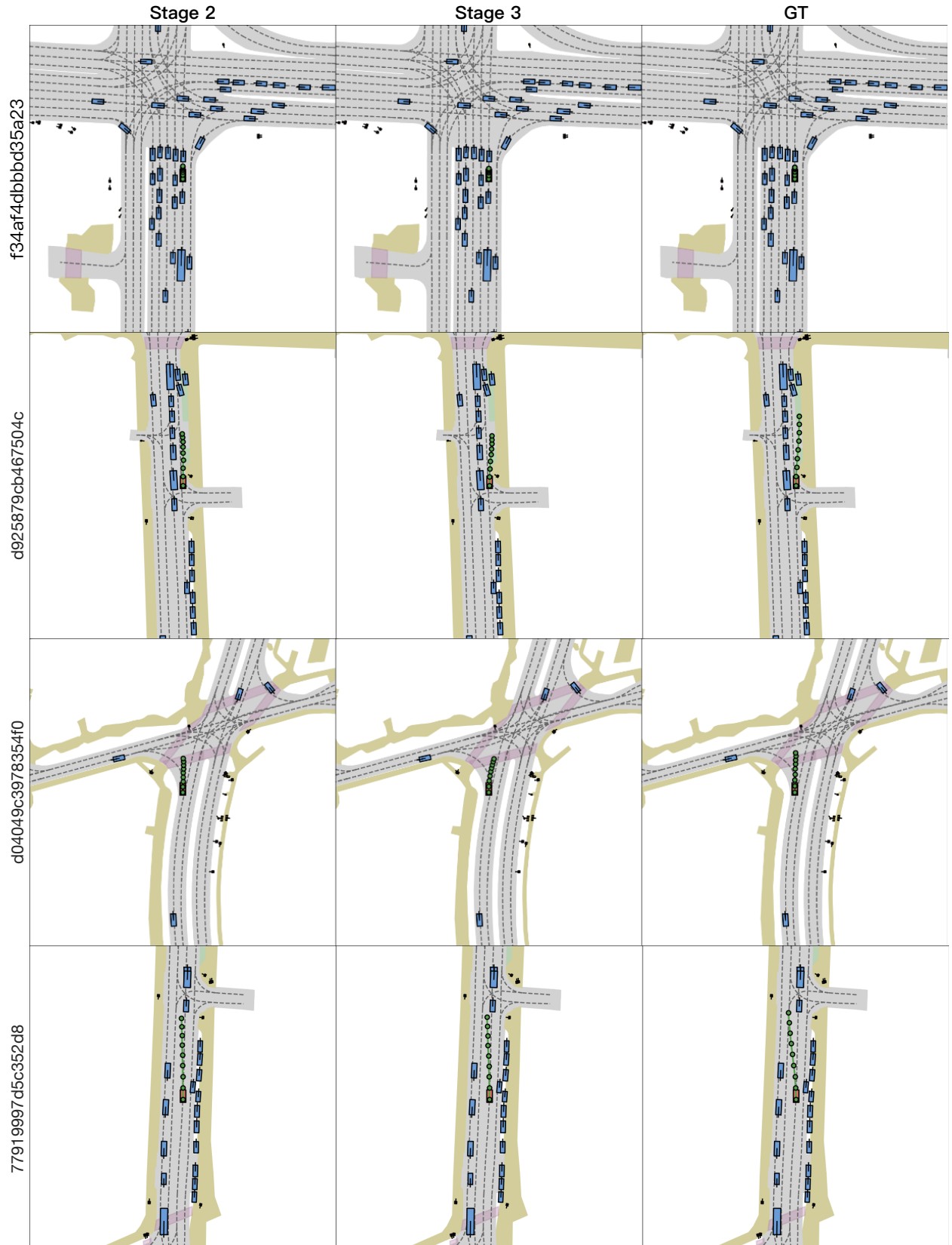

*Figure S4.* Visualization examples (cont.) of NAVSIM (Dauner et al., 2024). Left label: sample tokens. Top label: source of trajectory.

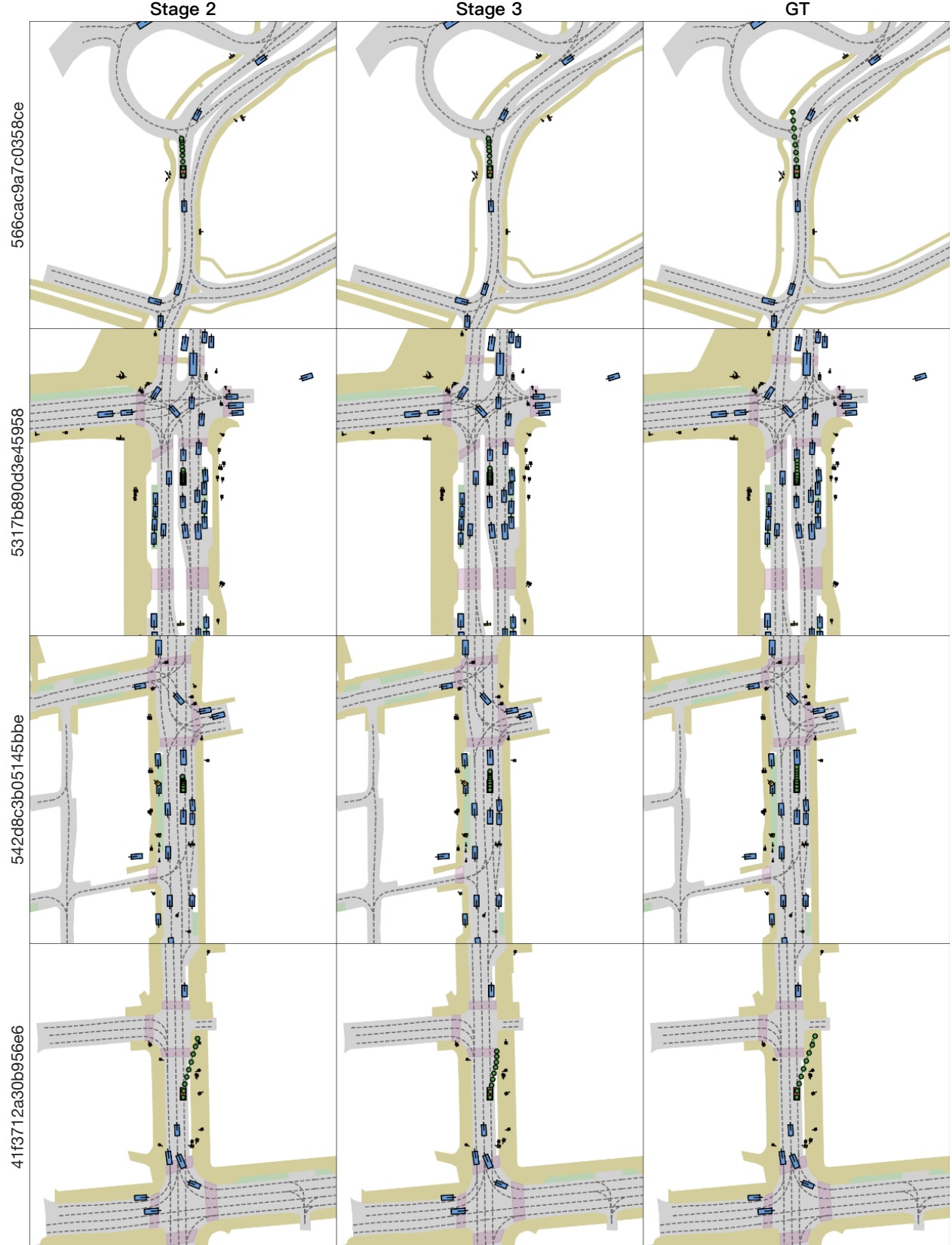

*Figure S5.* Visualization examples (cont.) of NAVSIM (Dauner et al., 2024). Left label: sample tokens. Top label: source of trajectory.

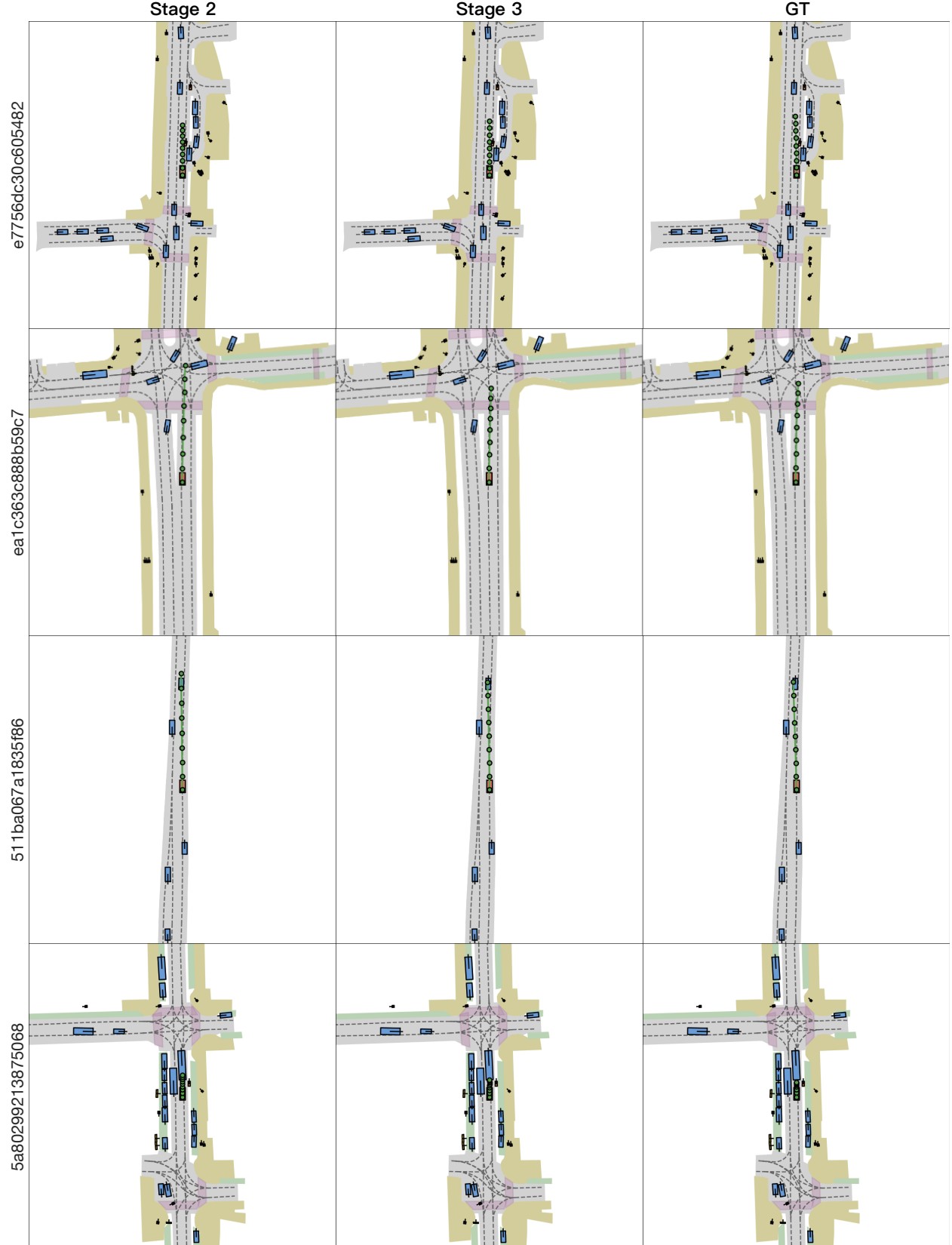

*Figure S6.* Visualization examples (cont.) of NAVSIM (Dauner et al., 2024). Left label: sample tokens. Top label: source of trajectory.

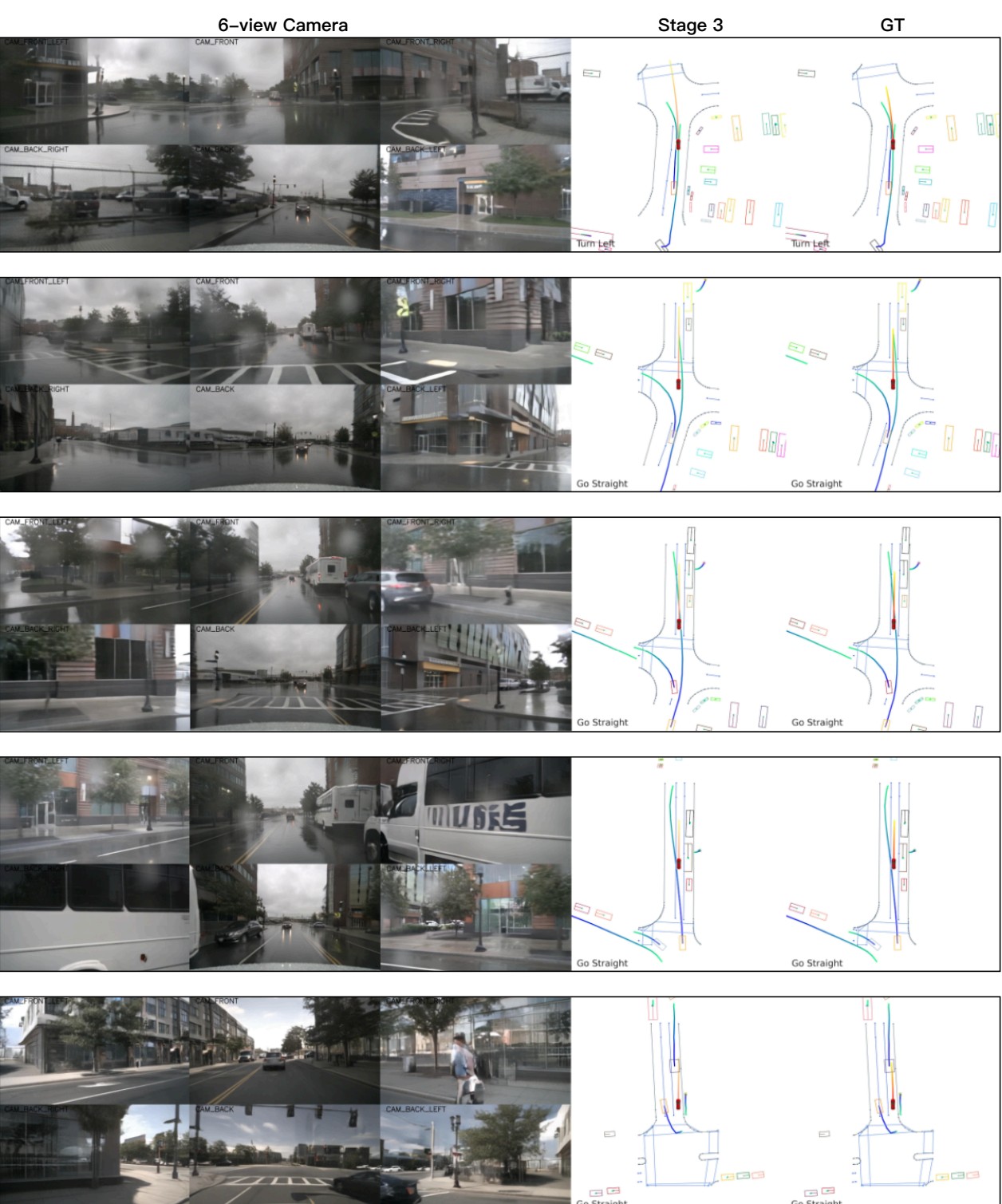

`

*Figure S7.* Visualization examples of nuScenes (Caesar et al., 2020) validation dataset. Top label: source of trajectory.

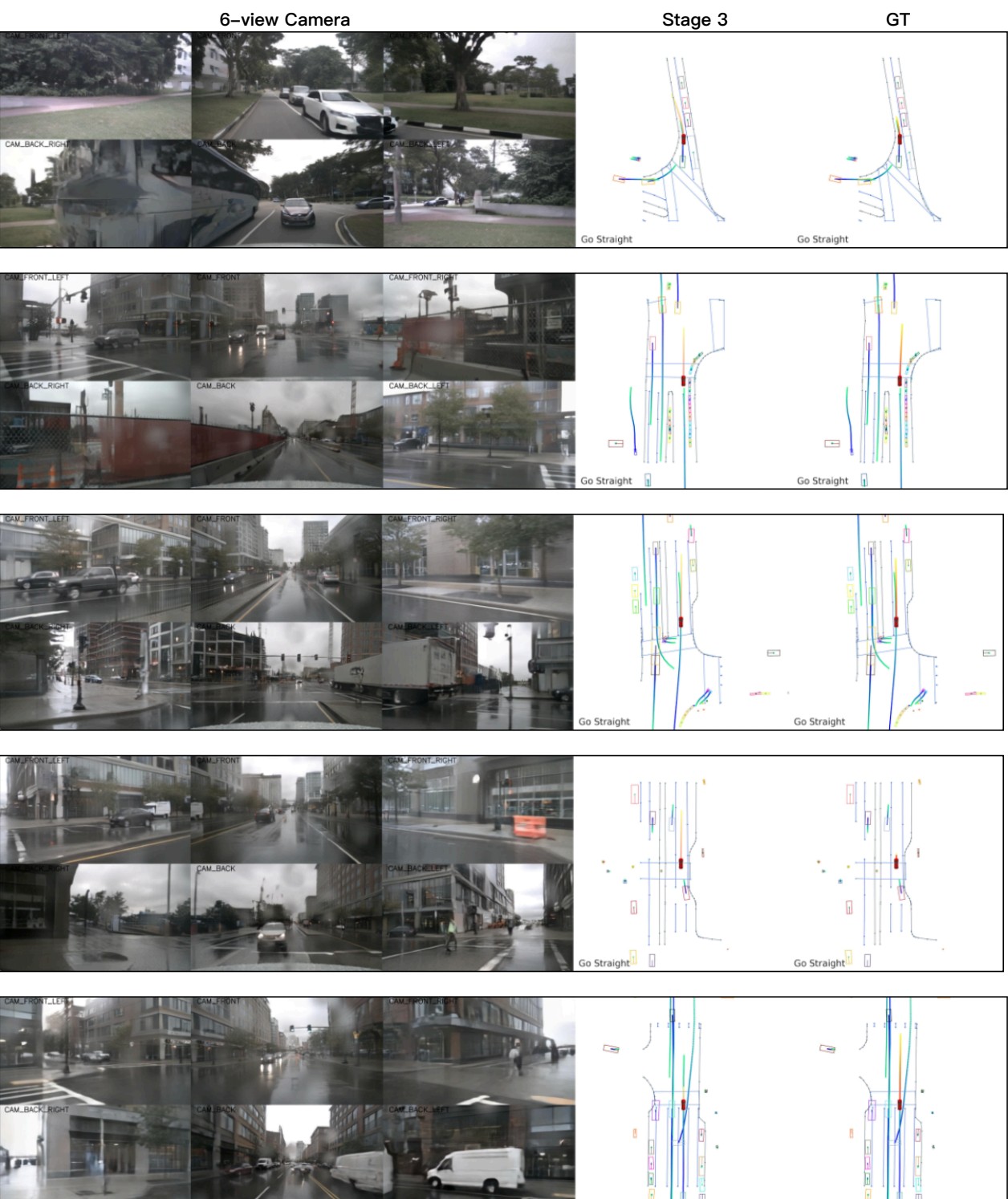

*Figure S8.* Visualization examples (cont.) of nuScenes (Caesar et al., 2020) validation dataset. Top label: source of trajectory.

