# OpenReview forum: "DriveWorld-VLA: Unified Latent-Space World Modeling with Vision–Language–Action for Autonomous Driving"
_ICML.cc/2026/Conference — ICML 2026 regular_

### Official Review · Reviewer_RbMc · 2026-03-02

**Soundness:** 3
**Presentation:** 3
**Significance:** 3
**Originality:** 3
**Overall Recommendation:** 4
**Confidence:** 3

**Summary:**

This paper proposed DriveWorldVLA, a three-stage VLA model that performs planning for autonomy tasks based on 2D image sequence and BEV data. The highlight of the model is the integration of world models and planning directly in latent space, utilizing spatial reasoning and simulators for future state prediction to enhance training. Some weaknesses include the over-reliance of BEV features, potential limitations of over-reliance on simulators and clean urban datasets, and hallucinations from both LLMs and diffusion models. Through thorough experiments, the model achieved sota performance on two datasets, leading me to believe the novelty and contribution of the paper.

**Compliance With Llm Reviewing Policy:**

Affirmed.

**Key Questions For Authors:**

1. Is the denoiser and diffusion model accurate and reliable enough? Do we have method to confirm this?
2. What dataset is used for training? Can you provide more details of the training process?
3. The debate of feature level vs. pixel level rollouts have been focused on efficiency rather than performance. Since you are already using expensive hardware, does using pixel level rollout improve the performance?

**Limitations:**

Yes.

**Strengths And Weaknesses:**

Strengths:
1. the integration of world models introduced more domain knowledge and reasoning capabilities to aid the decision-making process without more data.
2. the "what-if" situations could improve the model's reliability when it comes to edge cases in autonomous driving. The model might be able to provide more considered decisions rather than relying on data distribution.
3. the model achieved sota performance across two datasets.

Weakness:
1. the reward refinement process might introduce bias towards specific datasets, leading to credibility issues.
2. the model relies heavily on BEV data and 2D images; in stage 2, the model needs multi-view data to form BEV ground truth, which might not benefit the inference stage where such data is not available.
3. The model is not necessarily grounded to 3D geometries as the simulators used and the lack of 3D data might introduce additional unreliability. Furthermore, the hallucination of LLM and diffusion models are not explicitly handled.

---

> ### Author Rebuttal · Authors · 2026-03-31
>
> We appreciate the reviewer’s careful review.
>
> ***
>
> > `W1`: Bias from the Reward Refinement Process.
>
> We implement DriveWorld-VLA on different benchmarks, yet the reward optimization process is nearly identical, with only the trajectory scoring rules differing. For NAVSIM, we follow the scoring scheme in GTRS [1]. For nuScenes, Collision Rate serves as the sole optimization objective. **Thus, no data bias is introduced.**
>
> ---
> > `W2`: Train-Inference Inconsistency on BEV.
>
> This is a misunderstanding. During training, future BEV ground truth is only used as part of the supervision signal. In the inference stage, no supervision is required, and BEV features are generated directly.
>
> ---
> > `W3`: "Lack of 3D Data" and "Hallucination".
>
> * (1) **For "Lack of 3D Data"**: Many established works adopt the pure vision paradigm, such as Epona [2], ReCogDrive [3], DriveVLA-W0 [4], and LAW [5] listed in Table 1. None of these methods rely on LiDAR or other 3D data, and the experimental results of DriveWorld-VLA fully validate the reliability of this paradigm.
>
> * (2) **For "Hallucination"**: We have provided additional experiments in our responses to **Reviewer MkKx** `W1` and `W2` to verify the reliability of the VLM features. Meanwhile, the segmentation visualization results in `Q1` also demonstrate the stability of the diffusion model.
>
> ---
> > `Q1`: Denoiser Reliability.
>
> We provide an illustration of semantic segmentation to demonstrate the accuracy and reliability of our diffusion model: [Anonymous Link: https://i.postimg.cc/FFMV8X49/Drive-World-VLA-Rebuttal-vis.png](https://i.postimg.cc/FFMV8X49/Drive-World-VLA-Rebuttal-vis.png).
>
> According to our observation, the denoiser is reliable and produces semantically consistent features.
>
> Implementation: based on Stage 3, we freeze the parameters of the diffusion model and decode the generated features through a semantic segmentation head.
>
> ---
> > `Q2`: Details of the Training.
>
> We train DriveWorld-VLA on **nuScenes** and **NAVSIM** respectively. Beyond the implementation details presented in **Sec. 4.1**, additional settings are provided as follows:
>
> * (1) **For NAVSIM**: We implement the NAVSIM variant of DriveWorld-VLA with Python 3.8, PyTorch 2.0.1, and nuplan-devkit v1.2. No data augmentation is applied during training. Input images are normalized using the standard ImageNet statistics \(RGB mean: \[123.675, 116.280, 103.530\], standard deviation: \[58.395, 57.120, 57.375\]; the same applies hereinafter\). We use the AdamW optimizer with a weight decay of 0.0001. The image encoder is fine-tuned at $0.1\times$ the base learning rate, with a cosine annealing learning rate schedule.
>
> * (2) **For nuScenes**: We implement the nuScenes variant of DriveWorld-VLA with Python 3.9, PyTorch 2.1.0, and mmdetection3d v0.18.1. Standard 3D perception augmentations are adopted during training, including random horizontal flipping, global random scaling with a factor of 0.4, and random rotation and translation in the BEV view. Input images are normalized using the standard ImageNet statistics. The AdamW optimizer is used with a weight decay of 0.01 and a cosine annealing learning rate schedule. L2 norm gradient clipping is applied during training, with a maximum norm threshold of 35.
>
> ---
> > `Q3`: Feature Level Vs. Pixel Level Rollouts.
>
> Pixel-level rollouts preserve denser geometric and semantic details of the environment, which naturally improves model performance while incurring higher computational costs. The core of DriveWorld‑VLA lies in the mutual reinforcement between future generation and decision optimization. If one intends to use pixel-level rollouts to guide trajectory planning, it is still necessary to encode them into features or employ rule-based judges, such as those in Drive-WM [6]. Thus, we adopt the current design.
>
> ***
>
> ## Reference
> * [1] Generalized Trajectory Scoring for End-to-End Multimodal Planning (GTRS, CVPR 2025 Workshop)
> * [2] Epona: Autoregressive Diffusion World Model for Autonomous Driving (ICCV 2025)
> * [3] ReCogDrive: A Reinforced Cognitive Framework for End-to-End Autonomous Driving (ICLR 2026)
> * [4] DriveVLA-W0: World Models Amplify Data Scaling Law in Autonomous Driving (ICLR 2026)
> * [5] Enhancing End-to-End Autonomous Driving with Latent World Model (LAW, ICLR 2025)
> * [6] Driving into the Future: Multiview Visual Forecasting and Planning with World Model for Autonomous Driving (Drive-WM, CVPR 2024)

---

> > ### Author Rebuttal · Reviewer_RbMc · 2026-04-03
> >
> > My concerns are fully resolved.

---

> > > ### Author Response · Authors · 2026-04-03
> > >
> > > Thank you for your feedback!
> > >
> > > We would highly appreciate it if you could raise your rating.

---

### Official Review · Reviewer_mFBs · 2026-03-05

**Soundness:** 3
**Presentation:** 3
**Significance:** 3
**Originality:** 3
**Overall Recommendation:** 4
**Confidence:** 4

**Summary:**

This paper proposes a joint VLA and diffusion architecture. Hidden states from the final layer of the VLA and BEV features are fed into a DiT to generate future BEV features.
Training has three stages: (1) joint training; (2) training the BEV feature generator conditioned on future ground-truth actions; and (3) training the reward model and action head.
The three-stage training improves performance on the NAVSIM and nuScenes benchmarks.

**Compliance With Llm Reviewing Policy:**

Affirmed.

**Final Justification:**

My main concern has been resolved, so I maintain my recommendation.

**Key Questions For Authors:**

1. Since the world model predicts future BEV features encoded by BEVFormer, does this limit the potential of the world model?

2. In Table 5, why does “Non-progressive” perform worse than training without three-stage training (Table 4)?

3. In Section 3.1, how are the BEV features projected into the VLM space?

**Limitations:**

yes

**Strengths And Weaknesses:**

### Strengths

This paper explores how a world model (on BEV features) and a VLA planner can collaborate to improve driving performance.

The ablation studies demonstrate the necessity of incorporating three-stage training. The three-stage training is feasible and well motivated.

### Weaknesses

Using BEV features encoded by BEVFormer as input to the VLM may be unreasonable, or at least requires more careful design and justification.

---

> ### Author Rebuttal · Authors · 2026-03-31
>
> We are grateful for the reviewer's input.
>
> ***
>
> > `W1&Q1`: BEV Features as VLM Input and Output.
>
> We understand the reviewer’s concerns regarding this design. Combined with pioneering studies and theoretical analysis, we confirm that feeding BEV features into the VLM is reasonable and effective. **This design does not limit the potential of the world model; instead, it better meets the task requirements of autonomous driving scenarios.**
>
> * (1) This technical pipeline has been adopted and validated by multiple pioneering works in autonomous driving, and it is not our innovative design. Representative studies including HERMES[1] (ICCV 2025), BEV-InMLLM[2] (CVPR 2024), Talk2BEV[3] (ICRA 2024), ChatBEV[4] (arXiv 2025), and BEV-VLM[5] (arXiv 2025) all utilize this pipeline and demonstrate state-of-the-art performance.
> * (2) BEV features can effectively eliminate ambiguity caused by multi-view inputs and compress redundant image information. Meanwhile, they fully preserve scene geometric structures, road topological relationships, and semantic information. This well matches the long-sequence modeling capability and world commonsense reasoning needs of VLMs, and satisfies the strong demand for spatial consistency in world models.
> * (3) BEVFormer is not the only choice for BEV feature encoder, but rather the most commonly used one. We will explore more representation forms in future work.
>
> ---
> > `Q2`: "Non-progressive" Setting in Table 5.
>
> * (1) **The performance of the "Non-progressive" setting in Table 5 demonstrates the limitations of jointly optimizing the future feature generation task and the action decision optimization task.** According to the design logic of DriveWorld-VLA, the core of the second stage is to train an encoder that can accurately encode future features. The core of the third stage is to perform action learning using the future features generated in the second stage. This sequential design cannot be ignored. Only by first optimizing the generation task to obtain high-quality future features in Stage 2 can the action learning in Stage 3 achieve better performance.
> * (2) To address the concerns raised by **Reviewer MkKx** `W3` and **Reviewer mEFN** `W1` regarding the three-stage pipeline, we supplement the joint training of Stage 1 and Stage 2 based on Table 5, as shown in **Table R1**. **The performance drop further validates the claim in (1).**
>
> **Table R1**: Training Strategies based on Table 5.
> |Training Strategies|NC|DAC|TTC|C|EP|PDMS|
> |----|----|----|----|----|----|----|
> |Non-Progressive 1|97.8|94.5|93.5|99.9|75.6|83.6|
> |**Non-Progressive 2 (New)**|84.2|74.3|74.9|100.0|45.7|**52.5**|
> |Progressive|99.1|98.2|96.1|100.0|85.9|91.3|
> * (3) The "Non-progressive" result in Table 5 adopts different experimental settings from those in Table 4, and we have fully clarified this point in the current manuscript.
>
> ---
> > `Q3`: BEV Features Projection Pipeline.
>
> Full pipeline of BEV features is described as follows:
>
> First, multi-view images $\mathcal{I}_t$ are fed into BEVFormer to obtain BEV features $\mathcal{B}_t \in \mathbb{R}^{H\times W\times C}$.
> $\mathcal{B}_t$ captures semantic and geometric information of the world, where $H$ and $W$ represent the spatial size of BEV features, and $C$ denotes the channel dimension of BEV features.
> To align the channel dimension of BEV features with that of the VLM, we implement a projection layer via a $1\times 1$ convolution, transforming the shape of $\mathcal{B}_t$ into $H\times W\times C/2$.
> When invoking the VLM, the channel-compressed BEV features are flattened, and learnable positional embeddings are added. By this step, the BEV features are fully aligned into the VLM feature space.
> Subsequently, all features are processed by the VLM to generate the corresponding VL tokens, as shown in Eq. (1).
> Finally, we perform query operations between BEV features and VL tokens to obtain the final BEV tokens, as shown in Eq. (2).
>
> ***
>
> ## Reference
> * [1] HERMES: A Unified Self-Driving World Model for Simultaneous 3D Scene Understanding and Generation (ICCV 2025)
> * [2] Holistic Autonomous Driving Understanding by Bird's-Eye-View Injected Multi-Modal Large Models (BEV-InMLLM, CVPR 2024)
> * [3] Talk2BEV: Language-enhanced Bird’s-eye View Maps for Autonomous Driving (ICRA 2024)
> * [4] ChatBEV: A Visual Language Model that Understands BEV Maps (arXiv 2025)
> * [5] BEV-VLM: Trajectory Planning via Unified BEV Abstraction (arXiv 2025)
> * [6] End-to-End Driving with Online Trajectory Evaluation via BEV World Model (WoTE, ICCV 2025)
> * [7] Enhancing End-to-End Autonomous Driving with Latent World Model (LAW, ICLR 2025)

---

> > ### Author Rebuttal · Reviewer_mFBs · 2026-04-01
> >
> > Thanks for the reply. My main concern has been addressed.

---

> > > ### Author Response · Authors · 2026-04-02
> > >
> > > Thank you for your feedback!
> > >
> > > If your doubt has been resolved, we would appreciate it if you could raise your rating.

---

### Official Review · Reviewer_mEFN · 2026-03-09

**Soundness:** 3
**Presentation:** 3
**Significance:** 3
**Originality:** 2
**Overall Recommendation:** 4
**Confidence:** 4

**Summary:**

The paper introduces DriveWorld-VLA, a framework that unifies Vision-Language-Action (VLA) models and World Models (WM) within a shared latent space for autonomous driving. The authors identify a "structural bottleneck" in existing methods that either treat world models as external simulators or share features without enabling action-conditioned causal reasoning.

**Compliance With Llm Reviewing Policy:**

Affirmed.

**Final Justification:**

I am satified with the rebuttal and would like to raise the initial score.

**Key Questions For Authors:**

1. How does the "what-if" reasoning process (generating multiple latent rollouts via Euler sampling) impact the real-time inference speed (Hz) of the model?

2. To what extent does the reward-driven refinement in Stage 3 contribute to the final PDMS score compared to a model trained only through Stage 2?

3. How does the model perform in out-of-distribution scenarios (e.g., extreme weather or rare traffic maneuvers) that were not well-represented in the imitation learning data?

**Limitations:**

Yes.

The authors mention that DriveWorld-VLA cannot form closed-loop reasoning without the specific fine-tuning stages and discuss the transition from reactive to proactive planning.

**Strengths And Weaknesses:**

Strengths:

1. It utilizes a structured three-stage training process (Joint Training, Controllability Fine-tuning, and Reward-based Refinement) to stabilize the complex task of joint optimization. The use of a DiT-based architecture for action-conditioned flow-matching provides a rigorous mathematical framework for modeling future environmental evolution.

2. The paper clearly defines the limitations of current "disentangled" and "feature-sharing" paradigms. The transition to the proposed "unified" approach is logical, and the technical illustrations (Figures 2 and 3) effectively communicate the model's internal architecture.

3. It addresses a critical bottleneck in autonomous driving: the gap between reactive planning and long-horizon causal reasoning. Achieving state-of-the-art results on major benchmarks like NAVSIM (91.3 PDMS) and nuScenes (0.16 collision rate) demonstrates high practical utility for the industry.

Weaknesses:

1. While the paper claims to reduce reliance on dense supervision, it still requires heavy data inputs, including semantic BEV maps and expert trajectories. Furthermore, the complexity of the three-stage pipeline creates a potential "cascade of failure," where errors in the initial latent representation phase could significantly compromise the final policy.

2. There is a concern regarding computational efficiency. The process of performing prospective "what-if" rollouts using a Diffusion Transformer for multiple candidate actions is likely computationally expensive, which may hinder its deployment on real-time vehicle hardware where low latency is critical.

3. While the unification is novel, the individual building blocks (BEVFormer, InternVL, Diffusion Transformers) are well-established in the literature. The contribution is more of a sophisticated architectural synthesis rather than the invention of a new fundamental learning primitive.

---

> ### Author Rebuttal · Authors · 2026-03-31
>
> We appreciate the constructive feedback.
>
> ***
>
> > `W1`: "Heavy Data Inputs" and "Cascade of Failure".
>
> * (1) **For "Heavy Data Inputs"**: We only adopt the standard labels from public autonomous driving datasets. No additional supervision cost is introduced.
>
> * (2) **For "Cascade of Failure"**: The ablation results in Table 4 and Table 5 have verified the robustness of our three-stage pipeline. Table 4 demonstrates that the model does not suffer from performance degradation caused by error propagation. Table 5 proves that our training strategy is not obtained from empirical tuning. In addition, **Reviewer MkKx** `W3` raises a similar concern. We further supplement extra variants with joint training of Stage 1 and Stage 2 based on Table 5, as shown in **Table R1**. The significant performance drop further confirms that the three-stage pipeline is necessary and robust.
>
> **Table R1**: Training Strategies based on Table 5.
> |Training Strategies|NC|DAC|TTC|C|EP|PDMS|
> |----|----|----|----|----|----|----|
> |Non-Progressive 1|97.8|94.5|93.5|99.9|75.6|83.6|
> |**Non-Progressive 2 (New)**|84.2|74.3|74.9|100.0|45.7|**52.5**|
> |Progressive|99.1|98.2|96.1|100.0|85.9|91.3|
>
> ---
> > `W2`: Efficiency.
>
> We fully agree with the reviewer that computational efficiency is a critical issue that cannot be ignored. We provide the FPS results for reference in **Table R2**. The efficiency challenge commonly exists among methods with similar technical pipelines. The core focus of this work is to improve the planning performance.
>
> **Table R2**: Inference Speed. \* result from Ref.[3].
> |Method|Benchmark for Inference|Generation|Planning|FPS(Hz)|GPU Device|
> |----|----|----|----|----|----|
> |\*Vista[1]|nuScenes & Waymo|✓|✗|0.58|-|
> |\*DriveWorld[2]|nuScenes & OpenScene|✓|✗|0.25|-|
> |\*Orbis-FM[3]|nuPlan & Waymo|✓|✗|0.70|NVIDIA A100|
> |Epona[4]|NAVSIM & nuScenes|✓|✓|0.43|NVIDIA 4090|
> |**DriveWorld-VLA(Ours)**|NAVSIM & nuScenes|✓|✓|**1.16**|NVIDIA H20|
>
> ---
> > `W3`: Architectural Synthesis Vs. Primitive Invention.
>
> DriveWorld-VLA is a foundational pipeline, and many recent approaches, such as DriveVLA-W0[5], LAW[6], and ReCogDrive[7], follow a similar structure built on common learning primitives. A key innovation of DriveWorld-VLA is making shared VLM hidden states a practical and effective design through tightly integrated latent-space modeling and optimization.
>
> ---
> > `Q1`: Impact of "What-If" Reasoning on Inference Speed.
>
> We provide the time consumption and corresponding FPS under different DiT sampling steps in **Table R3**.
>
> **Table R3**: The time consumption of DiT sampling steps.
> |DiT Sampling Steps|DiT Time(s)|FPS(Hz)|
> |----|----|----|
> |**4 (used)**|**0.154**|**1.16**|
> |10|0.420|0.88|
> |25|0.901|0.62|
>
> ---
> > `Q2`: Contribution of Stage 3.
>
> **Table 4** clearly demonstrates the impact of reward-driven optimization in Stage 3 on planning performance:
>
> * (1) For NAVSIMv1, introducing Stage 3 brings a significant improvement of +1.8 in PDMS (from 89.5 to 91.3).
> * (2) For nuScenes, the CR@Avg. changes by -0.03 (from 0.19 to 0.16).
>
> ---
> > `Q3`: Performance of OOD Scenarios.
>
> We supplement the planning performance of noisy versions on two benchmarks to simulate extreme scenarios, as shown in **Table R4**. Ref.[8] provides 27 types of corruptions for nuScenes, named nuScenes-C. We focus on three extreme weather conditions including snow, rain, and fog, as well as motion blur, a typical motion-level corruption. These corruptions are further transferred to NAVSIM-C. **Our DriveWorld-VLA achieves the best robustness among all compared methods.**
>
> **Table R4**: Planning performance on NAVSIM-C (PDMS) and nuScenes-C (L2 and CR\@Avg. 3s). Noise function from Ref.[8].
> |Benchmark|Method|Clean|Motion Blur|Snow|Rain|Fog|
> |---|---|---|---|---|---|---|
> |NAVSIM-C(PDMS)|TransFuser[9]|84.0|65.8|72.3|69.2|73.0|
> ||**DriveWorld-VLA(Ours)**|**91.3**|**73.0**|**80.6**|**85.2**|**76.8**|
> |nuScenes-C\[8\]\(L2/CR\@Avg.3s\)|LAW[6]|0.65/0.34|0.74/0.56|0.92/0.44|1.00/0.54|0.68/0.55|
> ||**DriveWorld-VLA(Ours)**|**0.61/0.16**|**0.62/0.45**|**0.80/0.27**|**0.98/0.40**|**0.66/0.20**|
>
> ***
>
> ## Reference
> * [1] Vista: A Generalizable Driving World Model with High Fidelity and Versatile Controllability (NeurIPS 2024)
> * [2] DriveWorld: 4d Pre-trained Scene Understanding via World Models for Autonomous Driving (CVPR 2024)
> * [3] Orbis: Overcoming Challenges of Long-Horizon Prediction in Driving World Models (arXiv 2025)
> * [4] Epona: Autoregressive Diffusion World Model for Autonomous Driving (ICCV 2025)
> * [5] DriveVLA-W0: World Models Amplify Data Scaling Law in Autonomous Driving (ICLR 2026)
> * [6] Enhancing End-to-End Autonomous Driving with Latent World Model (LAW, ICLR 2025)
> * [7] ReCogDrive: A Reinforced Cognitive Framework for End-to-End Autonomous Driving (ICLR 2026)
> * [8] Benchmarking Robustness of 3D Object Detection to Common Corruptions (nuScenes-C, CVPR 2023)
> * [9] Multi-modal Fusion Transformer for End-to-End Autonomous Driving (TransFuser, CVPR 2021)

---

> > ### Author Rebuttal · Reviewer_mEFN · 2026-04-01
> >
> > The overall rebuttal is acceptable. For the inference speed, the report of different methods on different GPUs are not fair and reasonable. I tend to keep the current rating.

---

> > > ### Author Response · Authors · 2026-04-02
> > >
> > > Thank you again for your comments. We have further discussed the **FPS issue**.
> > >
> > > If we have fully addressed your concerns, we sincerely hope you can consider raising your rating.
> > >
> > > ***
> > >
> > > For a **FAIR FPS comparison**, we have added **Table R3**. We emphasize again that efficiency challenges are common among methods with similar technical pipelines. Even though DriveWorld-VLA achieves the highest FPS in Table R3, this does not imply that it has an advantage in efficiency. The core focus of the current work is to improve the performance of the planning task.
> > >
> > > **Table R3**: Inference Speed (updated version for fair comparison).
> > > |Method|Benchmark for Inference|Generation|Planning|FPS(Hz)|GPU Device|
> > > | ---- | ---- | ---- | ---- | ---- | ---- |
> > > |Vista[1]|nuScenes|✔|✗|0.57|NVIDIA H20|
> > > |Epona[4]|NAVSIM & nuScenes|✔|✔|0.40|NVIDIA H20|
> > > |DriveWorld-VLA(Ours)|NAVSIM & nuScenes|✔|✔|1.16|NVIDIA H20|
> > >
> > > ***
> > >
> > > **Potential further questions regarding Table R2 and Table R3:**
> > >
> > > 1. **Is Table R3 really FAIR?**
> > >
> > > We guarantee that Table R3 is sufficiently fair, for the following reasons:
> > > * We obtained the FPS values for Vista[1] and Epona[4] using the official GitHub code, checkpoints, and running instructions from their respective repositories, all under the same hardware environment (NVIDIA H20). We ensured that all models were inferred on a single GPU with a batch size of 1.
> > > * For different models, the code environments were isolated, with only the required Python dependencies for each method to avoid interference. All hyperparameters followed the best-performance settings reported in the original papers.
> > > * We excluded the time consumption of DataLoader and checkpoint loading for all methods, including DriveWorld-VLA.
> > > * We also aligned the benchmarks so that all compared methods now use the same benchmark as DriveWorld-VLA.
> > >
> > > 2. **How were the FPS values in Table R2 obtained?**
> > >
> > > * The FPS values of Vista[1], DriveWorld[2], and Orbis-FM[3] in Table R2 were all taken from Table 13 of Orbis[3]. Orbis[3] only explicitly stated that its own method was measured on an A100, but did not specify that Vista[1] and DriveWorld[2] were also tested on an A100. Moreover, the original papers of Vista[1] and DriveWorld[2] did not report their inference speeds. Therefore, their hardware information is marked as empty.
> > > * The FPS of Epona[4] in Table R2 was calculated from the data in Table 2 of its original paper.
> > >
> > > 3. **Why are DriveWorld[2] and Orbis-FM[3] not included in Table R3 while they are present in Table R2?**
> > >
> > > DriveWorld [2] does not provide an official code repository. Although Orbis provides an official code repository, its benchmark is not aligned with ours. Therefore, we do not report these two methods in Table R3.
> > >
> > > 4. **Why do the compared methods in Table R3 achieve lower FPS on better hardware than in Table R2?**
> > >
> > > * For Vista [1]: This result is indeed our actual measured value. Due to insufficient information provided in the original Orbis[3] paper, we cannot further speculate on the cause of this discrepancy.
> > > * For Epona [4]: We conjecture that the discrepancy arises from insufficient precision of the runtime reported in Table 2 of the original Epona paper. The original Epona paper clearly states that the setting for best performance uses `DiT sampling step = 100`, with the largest overhead coming from the VisDiT module. The paper reports a runtime of `∼2s`, which we used for our initial FPS calculation. However, after multiple tests, the average runtime of the VisDiT module on an H20 GPU is `2.171285s`, and the average time for one full inference is `2.523721s`.
> > > >Table 2 of Epona[4]:
> > > >>Table 2. Inference speed. We evaluate our inference speed for generating a 3-second trajectory and a 512×1024 image per module on a single NVIDIA 4090 GPU.
> > > >>| `DiT sampling steps` | MST    | TrajDiT | VisDiT |
> > > >>|--------------------|--------|---------|--------|
> > > >>| 10                 | ~0.02s | ~0.03s  | ~0.3s  |
> > > >>| 100                | ~0.02s | ~0.3s   | `~2s`    |
> > >
> > > ***
> > >
> > > ## Reference
> > > * [1] Vista: A Generalizable Driving World Model with High Fidelity and Versatile Controllability (NeurIPS 2024)
> > > * [2] DriveWorld: 4d Pre-trained Scene Understanding via World Models for Autonomous Driving (CVPR 2024)
> > > * [3] Orbis: Overcoming Challenges of Long-Horizon Prediction in Driving World Models (arXiv 2025)
> > > * [4] Epona: Autoregressive Diffusion World Model for Autonomous Driving (ICCV 2025)

---

### Official Review · Reviewer_MkKx · 2026-03-10

**Soundness:** 3
**Presentation:** 3
**Significance:** 3
**Originality:** 3
**Overall Recommendation:** 4
**Confidence:** 3

**Summary:**

The authors bring VLA-based understanding and decision-making together with World Model-based future imagination within a unified framework, and propose DriveWorld-VLA, where the VLA states are reused as a shared latent space for both decision-making and future imagination. Overall, the method shows strong performance across a range of benchmarks.

**Compliance With Llm Reviewing Policy:**

Affirmed.

**Final Justification:**

All of my concerns have been addressed, and I am inclined to raise my recommendation to Weak Accept.

**Key Questions For Authors:**

see Weaknesses

**Strengths And Weaknesses:**

### Stengths

1. The motivation figure and the visualization results are helpful.

2. The paper is clearly organized and easy to follow.



### Weaknesses

1. DriveWorld-VLA uses the VLM hidden states for semantic understanding, future imagination, and action prediction at the same time. However, just because these hidden states are trained with multiple objectives does not necessarily mean they form a truly unified representation or that they are causally sufficient for decision-making.

2. The model takes textual prompts as input, but it is not clear whether the final decision process is actually guided by language-level reasoning or logical constraints, unlike methods such as NavGPT-2 [1]. It would be helpful if the authors could clarify whether the text mainly serves as additional context, rather than as an explicit reasoning signal, and also discuss whether chain-of-thought-style supervision might further improve interpretability or decision consistency.

[1] Navgpt-2: Unleashing navigational reasoning capability for large vision-language models

3. The three-stage progressive training strategy seems reasonable, but in the current version it comes across more like an engineering trick than a clearly motivated high-level design choice. Since the overall performance appears to depend quite a lot on the training schedule, the ablation in Table 5 feels a bit limited, and it would be helpful to include more variants to better explain where the improvements actually come from.

4. The tokenizer design should be explained more clearly. The paper says that image and text tokenization follow InternVL, but the details of the BEV tokenizer and the action tokenizer are still  unclear -- for example, how the BEVFormer outputs are turned into BEV tokens, what the token format looks like, and whether the action tokenizer is borrowed from OpenVLA or implemented in a different way.

5. For the metrics where DriveWorld-VLA performs worse than previous methods in Tables 1 and 2, it would be helpful to provide more analysis of the underlying reasons. If there is not enough space in the main paper, this discussion could be moved to the appendix.

---

> ### Author Rebuttal · Authors · 2026-03-31
>
> We appreciate the reviewer’s valuable comments.
>
> ***
>
> > `W1`: Validity of Hidden State.
>
> We agree that it is challenging to optimize all tasks using shared VLM hidden states in a fully unified representation. A key innovation of DriveWorld-VLA is to render this design reasonable and effective:
>
> * (1) **For "Unified Representation"**: If the shared latent space is poorly constructed, downstream heads will favor task-specific features (e.g., BEV) over shared ones. To verify our shared latent space and rule out independent subtask optimization, we conducted VLM hidden state perturbation, see **Table R1**. Perturbing VLM hidden states significantly degrades Stage 1 and Stage 3 performance, proving shared features are essential.
>
> **Table R1**: Impact of VLM Perturbation. The perturbation is drawn from a Gaussian distribution $\mathcal{N}(0, 0.05)$.
> |Stage|NC|DAC|TTC|C|EP|PDMS|
> |----|----|----|----|----|----|----|
> |Stage 1|46.9|41.8|36.7|12.1|9.3|7.3|
> |Stage 3|46.2|40.0|36.0|12.2|8.8|6.9|
>
> * (2) **For "Causally Sufficient Decision-making"**: `W3` verifies that the three-stage pipeline follows a logical sequence and is sufficient to support decision-making.
>
> ---
> > `W2`: The Role of Text Modality.
>
> DriveWorld-VLA differs from NavGPT-2 in that it does not generate explicit step-by-step reasoning; instead, text prompts act as latent conditional constraints fused with visual features to directly guide actions:
>
> (1) **Table 6**: Following ReCogDrive[1], we conduct pre-training on autonomous driving VQA datasets. As shown in **Table 6 Line 1**, removing pre-training drops PDMS from 91.3 to 87.2, verifying its necessity.
>
> (2) **Text Masking Experiment**: We further add text masking experiments, where the text input is replaced with empty strings during inference, as shown in **Table R2**. Removing text inputs leads to a consistent performance drop, demonstrating that language provides important guidance for decision-making.
>
> **Table R2**: Text Masking.
> |Text Masking|NC|DAC|TTC|C|EP|PDMS|
> |----|----|----|----|----|----|----|
> |Yes|98.2|94.0|94.4|99.9|75.6|84.1|
> |No|99.1|98.2|96.1|100.0|85.9|91.3|
>
> ---
> > `W3`: Three-stage Pipeline and Table 5.
>
> We supplement additional experiments and theoretical analysis to prove that the three-stage pipeline is not obtained by empirical tuning, but necessary and robust.
>
> * (1) We add extra variant experiments with joint training of Stage 1 and Stage 2 based on Table 5, and observe significant performance drops, as shown in **Table R3**.
>
> **Table R3**: Training Strategies based on Table 5.
> |Training Strategies|NC|DAC|TTC|C|EP|PDMS|
> |----|----|----|----|----|----|----|
> |Non-Progressive 1|97.8|94.5|93.5|99.9|75.6|83.6|
> |**Non-Progressive 2 (New)**|84.2|74.3|74.9|100.0|45.7|**52.5**|
> |Progressive|99.1|98.2|96.1|100.0|85.9|91.3|
>
> * (2) **The three-stage pipeline is derived from the objective requirements of our model architecture.** A well-trained encoder for ground-truth future BEV features is mandatory, so Stage 1 builds scene perception to let Stage 2 converge. Stage 3 learns actions from Stage 2’s features. An unconverged generative model forces Stage 3 to optimize on noisy signals, causing severe performance drops. This is confirmed by the non-progressive strategies in Table R3.
>
> ---
> > `W4`: Details of Tokenizer.
>
> We provide detailed illustrations of the BEV Tokenizer and Action Tokenizer as follows:
>
> * (1) **BEV Tokenizer**: First, multi-view images $\mathcal{I}_t$ are fed into BEVFormer to obtain BEV features $\mathcal{B}_t \in \mathbb{R}^{H\times W\times C}$. To align the channel dimension with that of the VLM, we adopt a $1\times 1$ convolutional projection layer to reshape $\mathcal{B}_t$ into $H\times W\times C/2$. For VLM input, the compressed BEV features are flattened and added with learnable positional embeddings. All aligned features are then processed by the VLM to generate VL tokens (see Eq. (1)). Finally, BEV tokens are obtained by querying between BEV features and VL tokens (see Eq. (2)).
> * (2) **Action Tokenizer**: The historical action sequence is converted into a text-form list, concatenated with the textual prompt, and encoded via the tokenizer.
>
> ---
> > `W5`: Analysis of Non-SOTA Metrics in Table 1 and Table 2.
>
> We provide brief analyses for key non-SOTA metrics in Tables 1 and 2 as follows.
>
> * (1) PDMS and EPDMS are comprehensive metrics. Fluctuations in individual subindices do not reflect overall performance.
> * (2) For DAC and TTC in NAVSIMv1:Our long-term planning focus leads to minor DAC/TTC trade-offs for safety.
> * (3) For EC in NAVSIMv2: Future trajectory prediction introduces minor motion jitter that reduces EC.
>
> ***
>
> ## Reference
> * [1] ReCogDrive: A Reinforced Cognitive Framework for End-to-End Autonomous Driving (ICLR 2026)

---

> > ### Author Rebuttal · Reviewer_MkKx · 2026-04-03
> >
> > Thank you to the authors for the detailed response. All of my concerns have been addressed, and I am inclined to raise my recommendation to Weak Accept.

---

> > > ### Author Response · Authors · 2026-04-03
> > >
> > > Thank you very much for your feedback and for upgrading your rating!

---

### Decision · Program_Chairs · 2026-04-30

**Decision:**

Accept (regular)

**Comment:**

After rebuttal, this submission received all four weak accept, so there is a consensus of accept for this submission from the reviewers. AC has checked the submission, the reviews, the rebuttal, and the discussion, and agreed with the consensus, thus accept is recommended. The core strength identified by all reviewers is the unified latent-space architecture. Unlike existing paradigms that treat world models as external modules or simple feature-sharers, DriveWorld-VLA integrates VLA and world models at the representation level, thus it is a good contribution to the community.